**Phytoplankton community structure in the VAHINE mesocosm experiment**

Leblanc[1], Karine, Cornet[1], Véronique, Caffin[1], Mathieu, Rodier[2], Martine, Desnues[2], Anne, Berthelot[1], Hugo, Turk-Kubo[3], Kendra, Heliou[2], Jules.

[1] Aix Marseille Univ, CNRS/INSU, Univ Toulon, IRD, Mediterranean Institute of Oceanography (MIO) UM110, 13288, Marseille, France

[2] Mediterranean Institute of Oceanography (MIO) – IRD/CNRS/Aix-Marseille Univ, IRD Nouméa, 101 Promenade R. Laroque, BPA5, 98848, Nouméa CEDEX, New Caledonia

[3] Ocean Sciences Department, University of California, Santa Cruz, 1156 High Street, Santa Cruz, CA, 95064, USA

Correspondinc author: K. Leblanc (karine.leblanc@univ-amu.fr)

## Abstract

The VAHINE mesocosm experiment was designed to trigger a diazotroph bloom and to follow the subsequent transfer of diazotroph derived nitrogen (DDN) in the rest of the foodweb. Three mesocosms (50 $m^3$) located inside the Nouméa lagoon (New Caledonia, South West Pacific) were enriched with dissolved inorganic phosphorus (DIP) in order to promote $N_2$ fixation in these Low Nutrient Low Chlorophyll (LNLC) waters. Initially, the diazotrophic community was dominated by diatom diazotroph associations (DDAs), mainly by *Rhizosolenia/Richelia intracellularis*, and by *Trichodesmium,* which fueled enough DDN to sustain the growth of other diverse diatom species and *Synechococcus* populations that were well adapted to limiting DIP-levels. After DIP fertilization (1 µM) on day 4, an initial lag time of 10 days was necessary for the mesocosm ecosystems to start building up biomass. Yet changes in community structure were already observed during this first period, with a significant drop of both *Synechococcus* and diatom populations, while *Prochlorococcus* benefited from DIP-addition. At the end of this first period, corresponding to when most added DIP was consumed, the diazotroph community changed drastically and became dominated by *Cyanothece*-like (UCYN-C) populations, which were accompanied by a monospecific bloom of the diatom *Cylindrotheca closterium*. During the second period, biomass increased sharply together with primary production and $N_2$ fixation fluxes near tripled. Diatom populations, as well as *Synechococcus* and nano-phytoeukaryotes showed a re-increase towards the end of the experiment, showing efficient transfer of DDN to non diazotrophic phytoplankton.

## 1. Introduction

Atmospheric dinitrogen ($N_2$) fixation by marine planktonic diazotrophic organisms is a major source of new N to the ocean, and this process is particularly important in sustaining primary productivity in oligotrophic N-limited environments at low latitudes (Capone et al., 2005). On a global scale, $N_2$-fixation estimates converge around $140 \pm 50$ Tg N $y^{-1}$ (Gruber, 2004). The increase in primary productivity through diazotroph derived nitrogen (DDN) has been shown to increase carbon (C) export to depth (White et al., 2013). Diazotrophs have also been seen to contribute directly to C export (Subramaniam et al., 2008; Karl et al., 2012) and together these processes are capable of significantly impacting the biological C pump (Dore et al., 2008; Karl et al., 2012). A wide variety of autotrophic organisms are able to fix atmospheric $N_2$, from picoplanktonic and nanoplanktonic sized unicellular cyanobacteria (termed UCYN) to the heterocyst diazotrophs in symbiotic association with diatoms (DDAs) and to the larger filamentous colonies of *Trichodesmium*. Each group possesses different growth and $N_2$-fixation potential uptake rates and responds differently to environmental factors, depending on their ecological niches.

If $N_2$-fixation rates are routinely measured in the oligotrophic ocean, much less is known about which organisms contribute to this process as well as the fate of this newly fixed $N_2$ in the planktonic community. The VAHINE (VAriability of vertical and tropHIc transfer of fixed $N_2$ in the south wEst Pacific) mesocosms experiment was designed to address this particular issue, and to determine the primary routes of transfer of DDN along the planktonic food web. This project aimed at following the dynamics of a diazotroph bloom and investigate the evolution of the rest of the planktonic community (heterotrophic prokaryotes, pico-, nano-micro-phytoplankton and zooplankton) during this bloom event in order to determine whether the DDN rather benefited the classical food web or the microbial loop, as well as following the evolution of fluxes and stocks of biogenic elements. Finally, the VAHINE experiment was designed to determine whether a diazotroph bloom would increase the C export fluxes to depth.

Due to inherent logistical difficulties in answering these questions, that is to follow a naturally occurring diazotroph bloom in the open ocean and quantify the fate of DDN as well as C transfer to depth, a new approach involving mesocosms deployment was carried out in this project. A set of three replicate large-volume (ca. 50 $m^3$) mesocosms equipped with sediment traps at their bottom end were deployed in a protected area of the Nouméa lagoon in New Caledonia (South West Pacific), a site known for its warm oligotrophic waters favorable to

recurrent *Trichodesmium* blooms (Rodier and Le Borgne, 2010) and characterized by high $N_2$-fixation rates (Bonnet et al., *under rev*.).

Lagoon waters in Nouméa are known to be primarily N-limited (Jacquet et al., 2006; Torréton et al., 2010), which would favor the growth of diazotrophic organisms, but DIP availability was also suggested to exert the ultimate control on N-input by $N_2$ fixation in the western side of the South Pacific Ocean (Moutin et al., 2005; 2008). After isolation of the water column inside the mesocosms, DIP was added to each mesocosm in order to stimulate a diazotroph bloom event. The VAHINE experiment successfully allowed to follow a 2-phase diazotroph succession, associated to some of the highest $N_2$-fixation rates measured in the South West Pacific and composed of a succession of various diazotrophic organisms: DDAs were abundant during the first half (P1) of the experiment (up to day 14), while unicellular $N_2$-fixing cyanobacteria from Group C (UCYN-C) dominated the diazotroph community during the second half (P2) of the experiment (days 15 to 23) (described in details in Turk-Kubo et al., 2015). In support of the other main results presented in this VAHINE special issue, this paper presents the evolution of the phytoplanktonic community structure during this experiment.

## 2. Material and Methods

### 2.1. Mesocosms

Three large volume mesocosms were deployed in an LNLC ecosystem at the entrance of the Nouméa lagoon (New Caledonia) located 28 km off the coast (22°29.1'S– 166°26.9'E) in 25 m deep waters (Fig. 1). This system is under the influence of oceanic waters coming from the South through the open shelf, which then exit the lagoon, pushed by trade winds and tidal currents through various openings of the barrier reef (Ouillon et al., 2010). The mesocosms consisted of three enclosed polyethylene and vinyl acetate bags equipped with sediment traps at bottom. The mesocosms approximate height was 15 m, with an opening of 4.15 $m^2$ and a total volume of ca. 50 $m^3$ (see Guieu et al., 2010 and Bonnet et al., 2016 for full technical description of the mesocosms). Mesocosms were deployed on January 12[th] 2013 by scuba divers and left opened to stabilize the in-bag water column for 24h. Mesocosms were enclosed the following day (day 1), and the experiment was carried out between January 13[th] and February 4[th] for 23 days. In order to alleviate potential DIP-limitation of diazotrophic organisms, the three mesocosms were homogeneously fertilized with 0.8 µM DIP on the evening of day 4 (see Bonnet et al., 2016 for details), marking the start of P1 (P0 corresponding

to the period prior to fertilization between day 1 and 4).

Sampling occurred every day at 7 am at three selected depths (1, 6 and 12 m) in each mesocosm (hereafter called M1, M2 and M3) from a platform moored next to them and water was collected in large 50 L carboys using a Teflon pump connected to PVC tubing. To ensure quick processing of samples, the carboys were immediately transferred to the R/V *Alis* moored 0.5 nautical mile from the mesocosms or to the inland laboratory setup for this occasion on the Amédée Island located 1 nautical mile off the mesocosms. The seawater surrounding the mesocosms (hereafter called lagoon waters) was sampled every day for the same parameters at the same three depths.

### 2.2. Sample collection and analyses methods

#### 2.2.1. Determination of chlorophyll a (Chl *a*) concentrations

Chlorophyll a (Chl *a*) concentrations were determined from 0.55 L water samples filtered onto 25 mm GF/F Whatman filters in the three mesocosms and outside at all sampling depths. *In situ* Chl *a* concentrations were determined by fluorometry after methanol extraction (Herbland et al., 1985), using a Turner Design fluorometer (module # 7200-040, Chl *a* extracted-acidification) calibrated with pure Chl *a* standard (Sigma).

#### 2.2.2. Determination of phycoerythrin (PE) concentrations

Water samples (4.5 L) were filtered onto 0.4 µm Nucleopore polycarbonate membrane filters (47 mm diameter) and immediately frozen in liquid nitrogen until analysis. In the laboratory, particles retained on the filter were resuspended in a 4 mL glycerol-phosphate mixture (50/50) after vigorous shaking, according to the *in vivo* method (Wyman, 1992). The PE fluorescence excitation spectra were recorded between 450 and 580 nm (emission fixed at 605 nm), using a Perkin Elmer LS55 spectrofluorometer and emission and excitation slit widths adjusted to 5 and 10 nm, respectively (Neveux et al. , 2009). As this method was developed for small *Synechococcus* cells, potential packaging effect could occur when measuring PE in larger cells such as *Trichodesmium*, but this remains to be documented. Estimates of phycoerythrin were obtained from the area below the fluorescence excitation curve, after filter blank subtraction. PE analyses were made only at 6 m-depth in the three mesocosms and in lagoon waters.

### 2.2.3.  Pico- and nano-phytoplankton enumeration by flow cytometry

Samples for flow cytometry were collected from each carboys corresponding to each mesocosms and lagoon waters at the three depths in 1.8 mL cryotubes, fixed with 200 µL of paraformaldehyde solution (2 % final concentration), flash frozen in liquid nitrogen, and stored at -80 °C. Flow cytometry analyses were carried out at the PRECYM flow cytometry platform (https://precym.mio.univ-amu.fr/) using standard flow cytometry protocols (Marie et al., 1999)

to enumerate phytoplankton. Samples were analyzed using a FACSCalibur (BD Biosciences, San Jose, CA). Briefly, samples were thawed at room temperature in the dark and homogenized. Just before analyses, 2 µm beads (Fluoresbrite YG, Polyscience) used as internal control to discriminate pico-plankton ($< 2 \, \mu m$) and nano-plankton ($> 2 \, \mu m$) populations, and Trucount$^{TM}$ beads (BD Biosciences) used to determine the volume analyzed, were added to each sample.

An estimation of the flow rate was calculated, weighing 3 tubes of samples before and after a 3 min run of the cytometer. The cell concentration was determined from both Trucount$^{TM}$ beads and flow rate measurements. The red fluorescence (670LP, related to Chl *a* content) was used as trigger signal and phytoplankton cells were characterized by 3 other optical signals: forward scatter (FSC, related to cell size), side scatter (SSC, related to cell structure), and the orange

fluorescence (580/30 nm, related to phycoerythrin content). Several clusters were resolved, *i.e.* *Synechococcus*, *Prochlorococcus*, pico- and nano-phytoeukaryotes. Another cluster appeared in the nano-planktonic class-size as a stretched cloud of highly dispersed red and orange fluorescence. Its positioning on the cytogram suggests that this cluster corresponded to a gradient of particules comprised between 2 and 20 µm, including a mix of live and dead cells

embedded in aggregates, mucus and maybe pollens (data not shown). All data were collected in log scale and stored in list mode using the CellQuest software (BD Biosciences). Data analysis was performed *a posteriori* using SUMMIT v4.3 software (Dako).

### 2.2.4.  Micro-phytoplankton enumeration by microscopy

Samples for micro-phytoplankton enumeration and identification were collected in each mesocosms at mid-depth (6 m) in 250 mL amber glass bottles and fixed with 5 mL neutralized formalin. Samples were stored in the dark and at 4°C until analysis. Diatoms and dinoflagellates were identified and counted in an Utermöhl chamber on a TE-2000 Nikon inverted microscope following Utermöhl, (1931). Sedimented volume was on average 140 ml but ranged between

100 and 180 ml depending on cell density.

### 2.2.5.  Biomass conversions

The different groups were converted to carbon biomass in order to present an estimated overview of the relative dynamics of each group. Pico-phytoplankton C was computed using average values compiled from a global ocean database, i.e. 60 fg C per *Prochlorococcus* cell, 255 fg C per *Synechococcus* cell and 1319 fg C per pico-phytoeukaryote cell (Buitenhuis et al., 2012). Nano-phytoeukaryotes were assimilated to a sphere of 4 µm diameter and converted to C using Verity et al. (1992), which was equivalent to 10 pg C per nano-phytoeukaryote cell. Unfortunately the cellular sizes were not measured during diatoms cell counts, thus diatoms were converted to C using average size data compiled for each species from a global ocean database (Leblanc et al., 2012). Results are therefore only meant to present relative evolution of diatoms during the main phases of the experiment and should be interpreted with caution. Finally, dinoflagellates were converted by assimilating them to a 30 µm sphere (which corresponds roughly to observations) and using Menden-Deuer and Lessard (2000).

### 2.2.6. Diazotroph abundances and targeted diazotroph community succession

The abundances of specific diazotrophic phylotypes was determined using quantitative polymerase chain reaction (qPCR) targeting a component of the nitrogenase gene (*nifH*) associated to nine diazotrophic phylotypes. Briefly, 50 L water samples were collected at each depth each day and filtered through 25 mm 0.2 µm Supor® filters (Millipore, Billarica, CA), using peristaltic pumping. Filters were flash-frozen in liquid nitrogen, then stored at -80°C and shipped on dry ice back to laboratory. Diazotroph phylotypes targeted were regrouped for this study into the unicellular groups A (UCYN-A1+UCYN-A2), B (UCYN-B) and C (UCYN-C), the gamma proteobacteria (γ-247744A11), the colonial filamentous non heterocyst-forming cyanobacteria (*Trichodesmium*) and last the heterocyst-forming diazotrophic symbionts associated with diatoms (DDAs). These include the symbionts *Richelia* sp. associated to *Rhizosolenia* species (het-1), *Hemiaulus* species (het-2) and the symbionts *Calothrix* sp. associated to *Chaetoceros* species (het-3). Results are presented in number of *nifH* copies $L^{-1}$ for each phylotype and cannot easily be converted to cellular abundance because there is very little information about the number of *nifH* copies per genome for each diazotroph target. For a full description of the methods used, refer to Turk-Kubo et al. (2015).

### 2.2.7. Nutrients

Samples for dissolved inorganic phosphorus (DIP), nitrate ($NO_3^-$) and total nitrogen (TN)

concentrations were collected in 40 mL glass bottles and stored at -20°C before analysis. Concentrations were determined using a segmented flow analyzer according to Aminot and Kérouel (2007). The detection limit was 0.01 µM for $NO_3^-$ and 0.005 µM for DIP respectively. Dissolved Organic Nitrogen (DON) was derived after substracting PON measurements from TN measurements, with a precision of 0.5 µM. PON samples were collected by filtering 1.2 L on precombusted (450°C 4h) and acid washed (HCl 10%) GF/F filters and analyzed according to Pujo-Pay and Raimbault (1994), with a detection limit of 0.06 µM for PON. Samples for $NH_4^+$ were collected in 40 mL glass bottles and analyzed by the fluorescence method according to Holmes et al. (1999) on a trilogy fluorometer. The detection limit was 0.01 µM.

### 2.2.8. Nitrogen fixation rates

Samples for nitrogen fixation measurements were collected in 4.5 L polycarbonate bottles and amended with $^{15}N_2$-enriched water according to Mohr et al. (2010). After 24h, samples were filtered on combusted (450°C, 4h) GF/F filters and stored at -20°C. Filters were then dried at 60°C for 24h prior to analysis using a Delta plus mass spectrometer (Thermo Fisher Scientific) coupled with an elemental analyzer (Flash EA, Thermo Fisher Scientific) for PON concentrations and PON $^{15}N$ enrichment determinations. The $^{15}N$ enrichment of the $N_2$ pool was measured by Membrane Inlet Mass Spectrometer according to Kana et al. (1994) and was found to be $2.4 \pm 0.2$ atom%. Fluxes were calculated according to the equation given in Montoya et al. (1996). For a full description of the method used, refer to Berthelot et al. (2015b).

### 2.2.9. Primary production rates

Samples for primary production were collected in 60 mL bottles and amended with $H^{14}CO_3^-$ and incubated for 3 to 4h on a mooring line close to the mesocosm at the same sampling depth. Samples were filtered on 0.2 µm polycarbonate filters and placed into scintillation vials with 250 µL of HCl 0.5 M. After 12h, 5 mL of ULTIMA Gold MV scintillation cocktail were added to each vial before counting on a Packard Tri-Carb ® 2100 TR scintillation counter. Primary productivity was calculated according to Moutin et al. (2002).

## 3. Results
### 3.1. Pigment distribution

Chlorophyll $a$ (Chl $a$) remained low (close to 0.2 µg L$^{-1}$) during the first 14 days of the experiment in all three mesocosms (Fig. 2) and similar to the lagoon waters. A significant increase, which was not observed outside of mesocosms was observed by day 15 in all 3 mesocosms, which characterizes the beginning of the second phase (P2). M1 and M2 behaved more closely with similar doubling in average concentrations to around 0.4 µg L$^{-1}$, but with a few peaks at higher concentrations (up to 0.7 µg L$^{-1}$ in M1, and up to 1.0 µg L$^{-1}$ in M2 on day 18). M3 showed a similar trend but with a higher increase of Chl $a$, with an average concentration of 0.7 µg L$^{-1}$ during P2, and a higher peak value of 1.4 µg L$^{-1}$ on day 21.

Following the DIP-addition, PE remained close to initial values in all three mesocosms (close to 0.1 µg L$^{-1}$) and lower than in lagoon waters until day 11 (Fig. 3). PE concentrations then increased to an average of 0.2 µg L$^{-1}$ in M1 and M2, with daily variations, but increased up to a higher average concentration (0.5 µg L$^{-1}$) in M3 during P2, with a peak value of 1.0 µg L$^{-1}$ on day 19. Lagoon concentrations remained lower than in M3, but slightly above M1 and M2 during the first 15 days (0.2 µg L$^{-1}$), and increased to parallel M3 concentrations between day 20 and 22.

### 3.2. Pico- & Nano-phytoplankton distribution

The numerically dominant organism in the phytoplankton community during the experiment was *Synechococcus* (Fig. 4), whose abundances ranged between 16 000 and 285 000 cells mL$^{-1}$ (min and max values for all mesocosms). During P0, abundances were initially high but decreased steadily right after mesocosm enclosure, in order to increase again only several days after DIP addition. The average concentrations during P1 (day 5-14) in all three mesocosms was 54 000 cells mL$^{-1}$ and it nearly doubled during P2 (day 15-23) to 116 500 cells mL$^{-1}$. *Synechoccocus* increased more strongly after day 15, but the largest increase was observed in M3 with a peak value on day 19 close to 285 000 cells ml$^{-1}$.

*Prochlorococcus* (Fig. 5) showed intermediate abundance values (<50 000 cells mL$^{-1}$). In contrast to *Synechococcus*, they were initially low during P0 (close to 10 000 cells mL$^{-1}$), but increased strongly right after DIP addition in the three mesocosms. Apart from this similar initial response, the evolution of this group was less reproducible between mesocosms, with different patterns observed. A net decrease was observed by day 10 in M1, while abundances peaked on this same day in M2 and were intermediate in M3. Overall abundances were almost twice as low in M1 (4 600 to 23 400 cells mL$^{-1}$) than in M2 and M3 (8 400-10 000 to 42 900-43 500 cells mL$^{-1}$). *Prochlorococcus* were more abundant towards the end of the experiment in

all mesocosms (>20 000 cells mL$^{-1}$), but with a much higher number in M3 on day 21 (>40 000 cells mL$^{-1}$).

Next in order of abundance, pico-phytoeukaryotes ranged between 500 and 7 500 cells mL$^{-1}$ on average (Fig. 6). They were present during P0 with abundances > 2 000 cells mL$^{-1}$ but decreased right after DIP addition. They remained in low abundances mainly until day 18, 265 where they increased in all mesocosms (up to > 3 000 cells mL$^{-1}$), but with twice as many cells (> 7 000 cells mL$^{-1}$) in M3 than in M1 and M2. Pico-phytoplankton showed more contrasted responses in the three mesocosms in the transition period between day 10 and 15, with an increase in abundance in M1, stable values in M2 and a decrease in M3.

Finally nano-phytoeukaryote abundances were between 400 and 3 700 cells mL$^{-1}$ (Fig. 270 7). They were generally lower during P0 (<1 000 cells mL$^{-1}$) and seemed to respond to DIP addition with a small increase in numbers following day 4. No clear pattern can be derived from their distribution during the experiment but for a general increase over the last few days (after day 20) and higher abundances in M3 (> 3 000 cells mL$^{-1}$), similarly to what was already observed for *Synechococcus* and pico-phytoeukaryotes.

### 3.3. Diatom community structure

The dominant micro-phytoplanktonic organisms during the experiment were diatoms (Fig. 8), which abundances ranged from 5 700 to 108 000 cells L$^{-1}$ in all mesocosms. They were initially high during P0, despite large variations between mesocosms already on day 2. They 280 seemed to increase slightly right after DIP addition on day 5 (except in M1) and then decreased during P1 in all mesocosms until day 10-11, when they again started to grow, building up to bloom values (100 000 cells L$^{-1}$) around day 15-16 in the three mesocosms (on average twice as large in M1 than in M2 and M3).

The diatom community was composed of a diverse assemblage, which changed 285 significantly over the course of the experiment. From day 2 to day 12, the diatom community structure was diverse but very reproducible between mesocosms, despite near triple differences in abundances. Diatoms were initially numerically dominated by *Chaetoceros* spp. (*Hyalochaete* and *Phaeoceros*), which together accounted for 25 to 36 % of the total diatom abundances (Fig. S1). *Chaetoceros* spp. remained the most abundant group a couple of days 290 longer in M3, until day 14. In this first period, *Leptocylindrus* sp. was the second next most abundant genera contributing to 21 to 33 % to total diatoms in average over the 9 first days, and decreased to 16 % from day 10 to 12, and then remained below 10 % until the end of the experiment. *Cerataulina* spp.'s abundance was third next in M1, with 15 and 12 % contribution

in the first 5 days, but was below 5 % in the other 2 mesocosms. *Bacteriastrum* spp.'s abundance
was third next in M3 with 12% over the first 7 days, while it remained below 5 % in the two
other mesocosms. Finally *Thalassionema* spp.'s contribution was close to 10 % over the first 7
days in all three mesocosms, and decreased strongly after day 7.

From day 10 to day 18-19, *Cylindrotheca closterium,* which was inferior to 2 % of diatoms
in the first few days, increased dramatically in all three mesocosms and represented between
33 and 86 % of the diatom abundances, even reaching between days 15 and 17 > 95 % of total
abundance. After days 18-19, their contribution decreased again in favor of *Navicula* spp.,
*Chaetoceros* spp., *Leptocylindrus* spp. and *Guinardia* spp.

### 3.4. Dinoflagellate distribution

Dinoflagellates average abundance over the experiment was ca. 3 000 cells $L^{-1}$, an order
of magnitude inferior to diatoms. Dinoflagellates varied from 1 000 to 11 700 cells $L^{-1}$ (min
and max values for all mesocosms) and increased slightly (by a factor of 1.3 on average over
the three mesocosms) between P1 and P2, but this increase was more pronounced in M3 on
days 16-17. Average dinoflagellate abundance was also 3 to 4 times lower in M1 (1400 cells $L^{-1}$) compared to M2 (3400 cells $L^{-1}$) and M3 (4 400 cells $L^{-1}$). The numerically dominant species
were from the *Gymnodinium*/*Gyrodinium* group.

### 3.5. Biomass distribution of the phytoplanktonic community

The main phytoplanktonic groups were converted to C biomass and averaged for each
day of sampling for all mesocosms and all depths (Fig. 10). Given the assumption used for C
conversion (see methods section), these figures are only meant to give a rough estimate of C
allocation between groups, yet it has the merit to immediately convey the weight contribution
of each group, otherwise difficult to infer from abundance numbers.

Diatoms were the main contributors to phytoplankton C biomass (66 %) during P0 (day
2), while *Synechococcus* was the second largest contributor (19 %), followed by nano-
phytoeukaryotes (9 %). Nano-phytoeukaryotes relative biomass showed the strongest increase
during P1 (from 9 to 17 %), followed by *Synechococcus* (from 19 to 23 %) and dinoflagellates
(from 2 to 8 %) while diatoms decreased (from 66 to 47 %). During P2, *Synechococcus*
continued to increase (to 28 %) and diatoms to decrease (to 40 %) while all other groups
remained fairly stable. The evolution of *Prochlorococcus* contribution to biomass was
negligible over the course of the experiment and remained below 2 % during the 3 periods.

### 3.6. Diazotroph community distribution

The evolution of the targeted diazotrophic community is presented as the number of *nifH* copies L$^{-1}$ and as averages over the two main periods for a general overview of their relative dynamics (Fig. 11), since the evolution of the diazotrophic community on a daily basis during the experiment is described at length in the companion paper of Turk-Kubo et al. (2015). Important differences were observed between P1 and P2. The diazotroph community was

initially dominated by the Het-1 group (*Richelia*/*Rhizosolenia* association) with 1.3x10$^5$ *nifH* gene copies L$^{-1}$, followed by *Trichodesmium* with 2.0x10$^4$ *nifH* copies L$^{-1}$ and by UCYN-A with 1.7 x10$^4$ *nifH* copies L$^{-1}$. The Het-2 group (*Richelia/Hemiaulus* association) was less abundant (9.5x10$^3$ *nifH* copies L$^{-1}$), while the other groups (UCYN-C, γ24774A11, UCYN-B, Het-3 in decreasing order) were negligible in abundance (0 to maximum1 300 *nifH* copies L$^{-1}$).

During P2, the four dominant groups (Het-1, *Trichodesmium*, UCYN-A, Het-2) all decreased in abundance by a factor X3.7, X1.5, X1.6 and X1.7 respectively. The two least abundant groups UCYN-B, Het-3) during P1 increased only by a few hundred *nifH* copies L$^{-1}$ (with maximum values of 600 *nifH* copies L$^{-1}$). On the other hand the UCYN-C group showed a drastic increase (more than 10-fold) from 1.3x10$^3$ *nifH* copies L$^{-1}$ (P1) to 1.2x10$^5$ *nifH* copies

L$^{-1}$ (P2).

## 4. Discussion

        Following the DIP addition on the evening of day 4, the VAHINE experiment was

characterized by two distinct phases regarding nutrient availability, primary and heterotrophic bacterial production fluxes (Berthelot et al., 2015b; Van Wambeke et al., 2016) and the dynamics of the diazotrophs community as identified by quantitative polymerase chain reaction (qPCR) in Turk-Kubo et al. (2015). This experiment successfully triggered the development of a large diazotroph community, evidenced by the measured N$_2$-fixation fluxes which were

among the highest ever reported for oceanic systems (Bonnet et al., 2016). The first 10 days following the DIP fertilization (P1) were dominated mainly by DDAs, coupled with average N$_2$ fixation rates over the three mesocosms of 10.1 $\pm$ 1.3 nmol N L$^{-1}$ d$^{-1}$, while the following 9 days (P2) were dominated by the unicellular N$_2$-fixing cyanobacteria from group C (UCYN-C) which resulted in a near tripling of average N$_2$-fixation rates (27.3 $\pm$ 1.0 nmol N L$^{-1}$ d$^{-1}$) and a

more moderate increase in primary productivity, which increased from 0.9 to 1.5 µmol C L$^{-1}$ d$^-$

[1] between P1 and P2 (Berthelot et al., 2015b) (Fig. 12). A concomitant strong increase in average Chl *a* concentrations was observed, which nearly tripled from 0.20 to 0.54 µg L$^{-1}$. Similar primary production and N$_2$-fixation rates as well as Chl *a* concentrations were observed in the lagoon waters and the mesocosms during P1, but all parameters clearly increased during the second period inside the mesocosm experiment (Fig. 12), reflecting a delayed effect (~10 days) on the planktonic community, presumably affected by the combination of both DIP addition and turbulence reduction due to the entrapment of the water column. Results are discussed following this two-phase characterization of the evolution of the biological compartment over the course of the experiment.

## 4.1. Initial phytoplankton community composition during P0 (day 0-4)

The experiment started in LNLC waters, characterized by low (< 50 nM) DIN and DIP concentrations (Fig. 13), moderate DSi (1.4 µM) and low Chl *a* (0.2 µg L$^{-1}$) (Berthelot et al., 2015b). Primary production was on average low (0.4 µmol C L$^{-1}$ d$^{-1}$) while nitrogen fixation was elevated (17 nmol N L$^{-1}$ d$^{-1}$).

Diatoms were an important part (> 50%) of the phytoplanktonic biomass over the first few days (Fig.10, 14). This was surprising given the highly oligotrophic nature of the water mass, but can be explained by the presence of microplanktonic diazotrophs which could have stimulated the growth of other diatoms by indirect transfer of DDN. The diazotroph community analyses indicate that DDAs (in particular the het-1 *Rhizosolenia bergonii*/*Richelia* association) were dominant at the beginning of the experiment, and that other diazotrophs such as *Trichodesmium* and UCYN-A were also present.

Within the pico-phytoplankton size-class, *Synechococcus* was the dominant organism, representing 85 % of the C biomass, while *Prochlorococcus* and pico-phytoeukaryotes represented 5 and 10 % respectively. This relative allocation of biomass between these three groups remained stable throughout the experiment with very little variations (SD < 4% on all groups). A previous study conducted in the Nouméa Lagoon waters, showed that *Synechococcus* was dominating over *Prochlorococcus* over most of the DIN range and that pico-phytoplankton remained a negligible component of this size-class, which is consistent with our findings (Jacquet et al., 2006). *Synechococcus* was the most abundant group initially and 16S data showed that these high abundances were maintained in the Nouméa lagoon but that they crashed in M1 after mesocosm closure and further DIP addition (Pfreundt et al., 2016a).

This was the case in all three mesocosms (Fig. 6). In our system, both DIN and DIP were low, and the competitive advantage held by *Synechococcus* could derive from their ability to replace phospho-lipids in their cell membrane by sulfolipids during P-limitation (Van Mooy et al., 2009). Even if other groups such as *Prochlorococcus*, UCYN-B (*Crocosphaera*), *Trichodesmium* and some diatom species are also able to perform the same replacement of membrane lipids to save on cellular P demand (Van Mooy et al., 2009), it seems that *Synechococcus* was the most efficient organism using this substitution metabolism to resist P-stress in our initial conditions (Pfreundt et al., 2016b). This group was probably also benefiting from DDN to circumvent *in situ* DIN limitation.

### 4.2. Phytoplankton community composition during P1 (day 5 - 14)

In the period following DIP addition, production fluxes remained very close to lagoon waters (Fig. 12) similarly to nutrient stocks (Fig. 13). Average primary production increased from 0.4 to 0.9 $\mu$mol C $L^{-1}$ $d^{-1}$ while $N_2$ fixation actually decreased slightly (from 17 to 10 nmol N $L^{-1}$ $d^{-1}$). DIP addition however impacted phytoplankton community of both diazotrophs and non diazotrophs which started to depart from initial conditions as described below.

The DIP addition did not seem to immediately alter the main diatom species distribution, which remained fairly stable from day 2 to day 9 (Figs. 8, S1). However, it seems that diatom concentrations, after a rapid surge on day 5 in M2 and M3 corresponding to higher DIP levels in these mesocosms compared to M1, decreased significantly until day 9 in all mesocosms (Fig. 8, 10). As a potential mechanism, the DIP addition could have stimulated diatom growth initially, which would have then pushed diatoms into N-limitation if DDN was not sufficient to sustain this sudden increase in growth, and could have resulted in this initial decline in cell numbers. Another hypothesis could be that the water column enclosure, by reducing turbulence or by increasing the predator-prey encounter occurrences, could have been detrimental to the accumulation of diatoms during the first few days.

Although DDAs dominated the diazotroph community during P1, they however did not dominate the diatom community as a whole. *Rhizosolenia bergonii* (associated to *R. intracellularis*) represented less than 2 % of the diatom biomass initially, i.e. before DIP addition and increased to only around 8 % of the diatom biomass during P1, which was otherwise dominated by the very large *Pseudosolenia calcar-avis* (10 to 90 $\mu$m diameter, 200 to 800 $\mu$m length), which has a disproportionate impact on biomass, despite very low cellular

abundance. This diatom is known as an "S-strategist" (Reynolds, 2006) i.e. it is a large, slow growing species adapted to high nutrient stress and high light level and is usually found in very small mixed layer depths and in very low nitrate waters. The rest of the dominating diatom flora during the first 9 days of P1 was also comprised of species known to thrive in warm nutrient poor waters such as *Cerataulina*, *Guinardia* and *Hemiaulus* genera, while the numerically dominant *Chaetoceros* and *Leptocylindrus* species were more ubiquitous and fast growing species (Brun et al., 2015). The relative high abundance of diatoms other than DDAs and S-strategists in these nutrient depleted waters at the beginning of the experiment could have been fueled by secondary release of DDN (Mulholland et al., 2004; Benavides et al., 2013; Berthelot et al., 2015a). During this first period, the majority (over 50 %) of $N_2$-fixation was associated to the $> 10$ µm size-fraction (Bonnet et al., 2015) and was most likely the product of both *Richelia* and *Trichodesmium* but it cannot be determined which of these groups contributed most to the nitrogen uptake flux and subsequent DDN release. Only one diatom cell count is available for the lagoon waters on day 16, but it confirms that diatom community structure outside the mesocosms remained similar to our initial assemblage, composed of *Chaetoceros*, *Leptocylindrus* and *Guinardia* as well as *Pseudosolenia calcar-avis*.

A significant shift was observed within the diatom community after a few days during the second half of P1. The numerically dominant group of *Chaetoceros* spp. was gradually replaced by the small pennate diatom *Cylindrotheca closterium*, initially present in all mesocosms but in low abundance. Despite this dramatic increase in cell numbers leading to a near monospecific bloom at the transition period between P1 and P2, the overall diatom biomass yet decreased due to the small size of this pennate species. Interestingly, the climax of *C. closterium* was synchronous with an increase in UCYN-C populations that was not observed in the lagoon waters at any time and where the diazotroph community remained characterized by an increasing amount of DDAs and decreasing UCYN-A populations (Turk-Kubo et al., 2015). Both UCYN-C and *C. closterium* populations closely followed the staggered decrease in DIP as well as a small increase in temperature in the three mesocosms (Berthelot et al., 2015) hinting to bottom-up control of these groups. This solitary pennate diatom is found worldwide in both pelagic and benthic environments. It is likely that its dominance occurred through a better adaptation to the shift in abiotic factors occurring in the mesocosms from day 5 and on, i.e. much higher DIP level, decreased turbulence, as well as small increases in both temperature and salinity around day 9 in all mesocosms, which were accentuated during P2 (Bonnet et al., 2015). In a previous study involving perturbation experiments in small volume microcosms

conducted in high latitude HNLC (High Nutrient Low Chlorophyll) waters in the Bering Sea and in New Zealand, it was also shown that a simple Zn addition was able to induce a very rapid shift from *Pseudo-nitzschia* spp. to *Cylindrotheca closterium* community through subtle interplays in both their affinity for this trace metal (Leblanc et al., 2005). It is likely that this rather small and lightly silicified species can be considered as an opportunist species with high growth rates, allowing it to rapidly outcompete other diatoms when abiotic conditions become favorable. In support of this hypothesis, massive developments of *C. closterium* have previously been observed during *Trichodesmium* blooms in the South West pacific as well as in the near shore waters of Goa in western India (Devassy et al., 1978; Bonnet et al., *under rev*.). One hypothesis for this recurrent co-occurrence of *C. closterium* with various diazotrophic groups would be that this diatom species has a better immediate affinity for DDN, probably in the form of $NH_4^+$, than other diatoms.

Several studies have previously demonstrated the development of diatoms as well as dinoflagellates following N release by *Trichodesmium* spp. (Devassy et al., 1978; Dore et al., 2008; Lenes and Heil, 2010; Chen et al., 2011; Bonnet et al., *under rev.*). In contrast, dinoflagellates here only showed a moderate increase towards the middle of the experiment (days 16-17) in M3 and an increase in the last few days in M2 but no clear trend could otherwise be detected (Fig. 9), and their biomass remained overall stable over the course of the experiment (Fig. 10). It is however possible that dinoflagellates growth may have been stimulated by DDN, but that their biomass was kept unchanged by subsequent grazing, or that their mixotrophic regime allowed them to exploit changes in the dissolved organic pool or go over to phagocytosis (Jeong et al. 2010).

In the pico-phytoplankton community, *Synechococcus* and pico-phytoeukaryotes exhibited very similar dynamics, with a distinct drop after DIP-addition and a re-increase with a higher degree of variability between the three mesocosms from the middle of P1 approximately (Figs. 4, 6, 13). A likely explanation would be that they started to benefit from DDN and increased growth rates again only once the UCYN-C population started to increase. On the other hand, *Prochlorococcus* clearly benefited from the DIP-addition (Fig. 5), with a strong increase in cell numbers in the beginning of P1, which yet only results in a relative increase of 1 % to phytoplankton biomass (Fig. 10). Nano-phytoeukaryotes, which were low initially increased right after DIP addition and continued to increase towards the end of P1 (Fig. 7) probably also thriving on DDN.

### 4.3. Phytoplankton community composition during P2 (day 15 to 23)

The second period of this mesocosm experiment showed major changes compared to P1. The introduced DIP was rapidly consumed during P1 (Fig. 13) allowing a strong build-up of biomass (Chl *a* and PE) together with a near tripling of $N_2$-fixation rates (27 nmol N $L^{-1}$ $d^{-1}$) which were significantly superior to lagoon values, which also increased but more moderately (Fig. 12). This evolution is likely due to the stimulation of a different diazotroph community inside the mesocosm, with higher $N_2$-fixation rates, which in turn increased DDN release and resulted in a larger consumption of all inorganic nutrients compared to outside waters (Fig. 13).

This second phase, corresponding approximately to the moment when DIP was completely consumed (to less than 0.1 µM), was characterized by an important shift in the diazotroph community. Differences between the mesocosms and lagoon waters were evidenced, the first being dominated by UCYN-C (*Cyanothece*), followed by het-1 (more abundant in M1) and *Trichodesmium* (more abundant in M3) while the latter were still dominated by DDAs, *Trichodesmium* and UCYN-A (Turk-Kubo et al., 2015). The UCYN-C cells (around 6 µm) grew in the mesocosms and rapidly achieved the highest *nifH* gene copies values for all diazotrophs during P2, while most other groups diminished (Fig. 11), most notably het-1 (Turk-Kubo et al., 2015).

At the transition between P1 and P2, the development of UCYN-C was paralleled by a drastic change in diatom community structure, which became almost monospecifically dominated by *C. closterium*. It seems however that this stimulating effect was not durable, as this *C. closterium* bloom started to crash rapidly (most significantly in M1), from day 19-20, which was accompanied by a shift in species distribution, with the return of the *Chaetoceros* spp. and the appearance of *Navicula* spp. One hypothesis regarding this sharp decline of *C. closterium* towards the end of the experiment could be top-down control by grazers, leading to a shift towards less palatable diatom species.

In the mesocosms, UCYN-C rapidly aggregated in the form of large aggregates (from 100-500 µm) and Berman-Frank et al. (2016) showed that UCYN-C abundances were positively correlated to transparent exopolymer particles (TEP) concentrations, which could hint to a direct production by these organisms. Moreover, *C. closterium* has also been associated to large mucilage aggregate formations in the Mediterranean (Najdek et al., 2005) and it is known to produce TEP under nutrient stress (Alcoverro et al., 2000). Thus, both the dominating diatom and the UCYN-C could have produced the TEP and/or TEP precursors leading to the formation

of these large aggregates in the mesocosms, which resulted in an important contribution (22% of POC) of UCYN-C to export in the mesocosm traps during the second phase (Bonnet et al., 2015).

Interestingly, *Synechococcus* increased again strongly during P2 (Fig. 14), showing its greater competitive advantage over other pico-phytoplankton groups in the P-limited and DDN rich environment by reducing its P cellular demand and use up the newly available DDN. This hypothesis was supported by gene expression dynamics from metatranscriptomic analysis which showed that *Synechococcus* (but not *Prochlorococcus*) was expressing genes for sulfo-

lipid biosynthesis proteins over the course of the experiment whenever it was abundant, and also increased transcript accumulation for $NH_4$ transporters towards the end of the experiment (Pfreundt et al., 2016b). Another competitive advantage is its mixotrophic character, as *Synechococcus* cells are also able to assimilate amino acids (Van Wambeke et al., 2016). Based on its genome, Palenik et al. (2003) have also shown that *Synechococcus* is clearly more

nutritionally versatile and a 'generalist' compared with its *Prochlorococcus* relatives, likely explaining its success in this experiment.

    In the last few days, the evolution of populations in M3 departed strongly from the other mesocosms, with higher primary productivity, $N_2$-fixing fluxes and biomass accumulation, originating from the larger development of *Synechococcus,* pico-phytoeukaryotes*,* but also

*Trichodesmium* populations, which may have been favored by the slower DIP decrease and the slightly higher salinities measured in this mesocosm compared to the other two. The PE signal showed a strong increase only in M3, and was likely mainly driven by this increase in *Synechococcus* and *Trichodesmium*, which were in much higher abundance in this mesocosm. It is likely that the PE accumulation was not so much correlated to the increase in UCYN-C, as

related *Cyanothece* strains did not show any PE signal in culture (comm. pers. Rodier) and because their contribution to biomass was rather small.

    The evolution of dinoflagellates, overall dominated by cells < 50 µm belonging to the *Gymnodinium*/*Gyrodinium* spp. mix, showed no distinct patterns between P1 and P2 (Figs. 9, 10, 14) and no reproducible trends between mesocosms, as detailed previously. Dinoflagellates

are comprised of autotrophs, heterotrophs as well as mixotrophs, which makes it difficult to relate their dynamics to bottom-up control factors, and is more likely reflecting the result of biological interactions with other groups.

## 5. Conclusion

The VAHINE mesocosm enclosure experiment and subsequent DIP-addition in coastal LNLC waters inside the Nouméa lagoon successfully triggered a succession in the diazotroph community that stimulated both primary production and exceptionally high $N_2$-fixation rates after a lag time of approximately 10 days compared to fluxes observed in the surrounding lagoon waters. A distinctly different planktonic community developed inside the mesocosms, which were generally well replicated despite slight timing and concentration variations of the different groups observed. A diverse diatom community was initially (P0) dominant in these nutrient limited waters, and was most likely fueled by DDN release by present DDAs (namely *Rhizosolenia/Richelia*), *Trichodesmium* and UCYN-A. *Synechococcus* was the other main component of phytoplankton and is known to hold a competitive advantage at limiting P levels with its ability to replace phospho-lipids by sulfo-lipids as well as use $NH_4^+$ from DDN.

After DIP addition, the average Chl *a* concentrations did not show any increase for another 10 days, yet shifts in the community structure were observed during this first period (P1). Both *Synechococcus* and pico-phytoeukaryotes populations dropped while *Prochlorococcus* clearly benefited from the sudden P availability. Diatoms, after an initial surge on the day following P addition without changes in community structure rapidly decreased and started to re-increase only after a week. Between day 11 and 15, a monospecific bloom of *C. closterium* developed, closely coupled to the apparition of UCYN-C populations, both following the staggered decrease in P-availability in the three mesocosms. The association of *C. closterium* blooms during other diazotroph bloom events has already been recorded in previous studies and indicates that this diatom species could be very efficient in using up DDN while P levels are still sufficient.

The second period (P2), when DIP was again depleted was defined by an important increase in Chl *a*, associated to increases in primary production and near tripled $N_2$-fixation rates. These changes were coupled to important shifts in the diazotroph community, which became dominated by UCYN-C, which rapidly aggregated. *Synechococcus*, diatoms and nano-phytoeukaryotes abundances re-increased towards the end of the experiment, revealing an efficient transfer of DDN to these groups, this time fueled by UCYN-C rather than by DDAs and *Trichodesmium.*

In conclusion, we show that the elevated $N_2$-fixation rates, stimulated by a DIP-fertilization in enclosed mesocosms in LNLC waters benefited the entire planktonic community with clear stimulation of both diazotrophic and non-diazotrophic groups mainly observed by

*Synechococcus* and diatom species other than DDAs, which has clear implications for the efficiency C export fueled by DDN.

*Author contributions.* S. Bonnet designed and executed the experiments. A.Desnues, H. Berthelot and M. Caffin contributed to sampling and analyses of flow cytometry and microphytoplankton sampling, M. Rodier sampled for pigments and flow cytometry, J. Héliou and M. Rodier analyzed pigment samples, V. Cornet analyzed microphytoplankton samples. K. Turk-Kubo analyzed the *nifH*-based abundances of targeted diazotroph taxa. K. Leblanc prepared the manuscript with input from all co-authors.

*Acknowledgements.* Funding for this research was provided by the Agence Nationale de la Recherche (ANR starting grant VAHINE ANR-13-JS06-0002), INSU-LEFE-CYBER program, GOPS and IRD. The authors thank the captain and crew of the R/V *Alis*. We also thank the SEOH diver's service from Nouméa, as well as the technical support and the divers of the IRD research center of Nouméa. We are grateful to the Regional Flow Cytometry Platform for Microbiology (PRECYM) of the Mediterranean Institute of Oceanography (MIO, Marseille, France) for the flow cytometry analyses. We gratefully acknowledge C. Guieu, J.-M. Grisoni and F. Louis from the Observatoire Océanologique de Villefranche-sur-mer (OOV) for the mesocosms design and their useful advice.

**Figure legend:**

Fig. 1: Location map of mesocosms deployment off Nouméa in New Caledonia.

Fig. 2: Total Chl *a* in µg L$^{-1}$ at each of the three depths (1, 6 and 12m) inside each mesocosm (M1, M2 and M3) and outside of mesocosms (OUT).

Fig. 3: Total phycoerythrin in µg L$^{-1}$ at the intermediate depth (6 m) inside each mesocosm and in the control area outside of mesocosms.

Fig. 4: *Synechococcus* in cells mL$^{-1}$ at each of the three depths (1, 6 and 12m) inside each mesocosm (M1, M2 and M3).

Fig. 5: *Prochlorococcus* in cells mL$^{-1}$ at each of the three depths (1, 6 and 12m) inside each mesocosm (M1, M2 and M3).

Fig. 6: Pico-eukaryotes in cells mL$^{-1}$ at each of the three depths (1, 6 and 12m) inside each mesocosm (M1, M2 and M3).

Fig. 7: Nano-eukaryotes in cells mL$^{-1}$ at each of the three depths (1, 6 and 12m) inside each mesocosm (M1, M2 and M3).

Fig. 8: Diatom genera/species abundance in cells L$^{-1}$ at the intermediate depth (6 m) in each mesocosm.

Fig. 9: Total dinoflagellate abundance (in cells $L^{-1}$) at the intermediate depth (6 m) inside each mesocosm.

Fig. 10 : Dynamics of the biomass of the main groups constituting phytoplankton communities in biomass (diazotrophs not included) over the course of the experiment for *Prochlorococcus* (PROC), *Synechococcus* (SYN), pico-phytoeukaryotes (PICO), nano-phytoeukaryotes

(NANO), diatoms (DIAT) and dinoflagellates (DINO).

Fig. 11: Boxplots of targeted diazotrophs groups in nifH gene copies $L^{-1}$ in the three mesocosms during the two periods P1 and P2.

Fig. 12: Boxplots of primary production (in µmol C $L^{-1}$ $d^{-1}$), $N_2$-fixation rates (in nmol N $L^{-1}$ $d^{-1}$) and Chl *a* concentrations (in µg $L^{-1}$) in the three mesocosms (top panels) and in the lagoon

waters (bottom panels) during the two periods P1 and P2.

Fig. 13: Boxplots of nutrients, DIP, $NO_3$, DON in µM and $NH_4^+$ (in nM) in the three mesocosms (top panels) and in the lagoon waters (bottom panels) during the two periods P1 and P2.

Fig. 14: Boxplots of the main phytoplanktonic groups in cells $L^{-1}$ in the three mesocosms during the two periods P1 and P2.

**Supplementary Figures:**

Fig. S1: Main diatom genera/species composition in % contribution at the intermediate depth (6 m) in each mesocosm.

Fig. S2: Average (±SD) contribution to C biomass of the main groups constituting

phytoplankton communities (diazotrophs not included) over the course of the experiment following the three periods P0, P1 and P2 for *Prochlorococcus* (PROC), *Synechococcus* (SYN), pico-phytoeukaryotes (PICO), nano-phytoeukaryotes (NANO), diatoms (DIAT) and dinoflagellates (DINO).

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

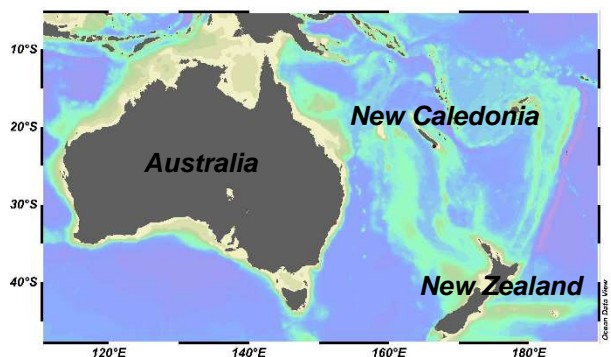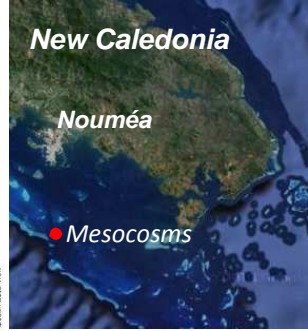

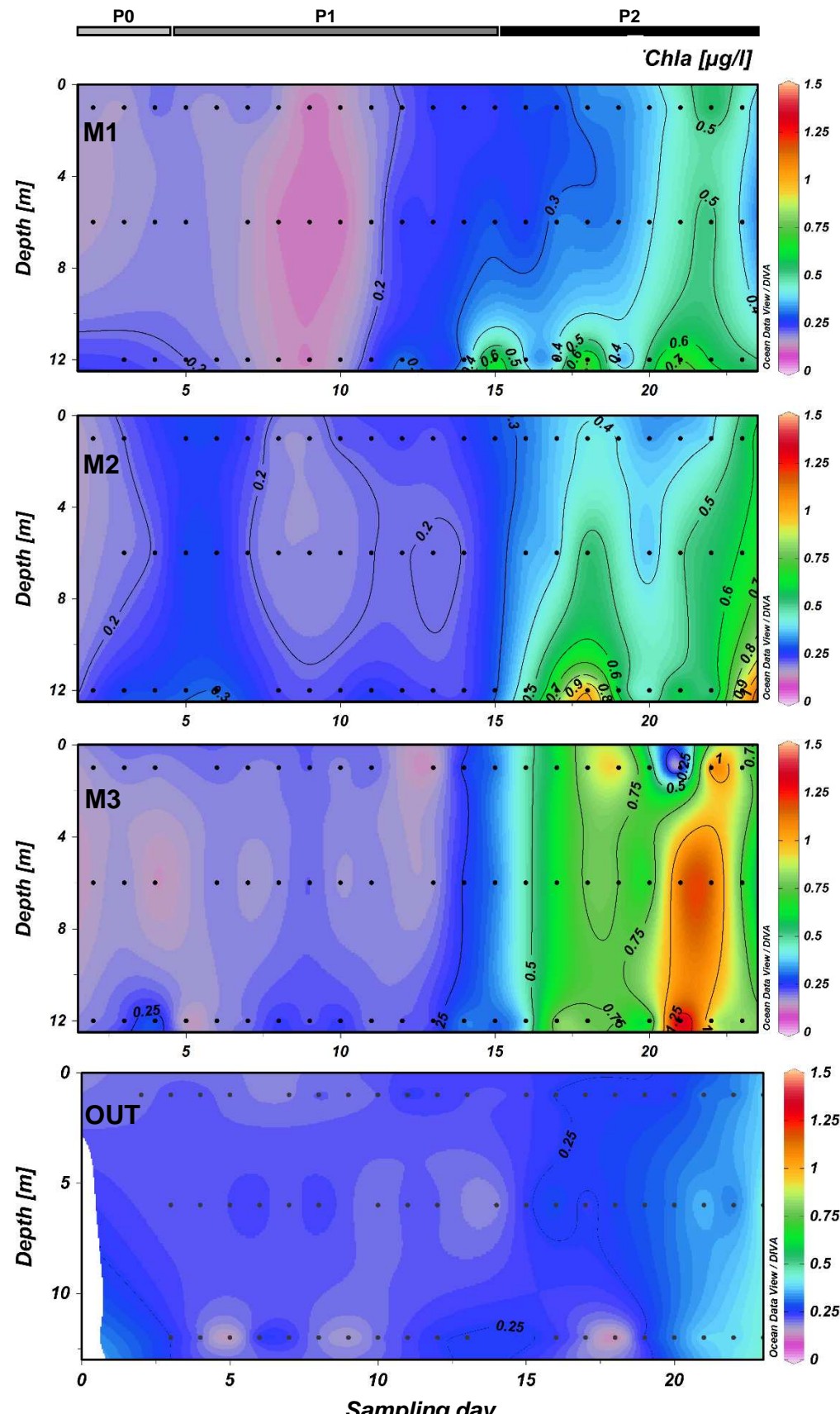

Figure 2 : Chl*a* in µg L$^{-1}$ at each of the three depths (1, 6 and 12m) inside each mesocosm (M1, M2 and M3) and outside of mesocosms (OUT).

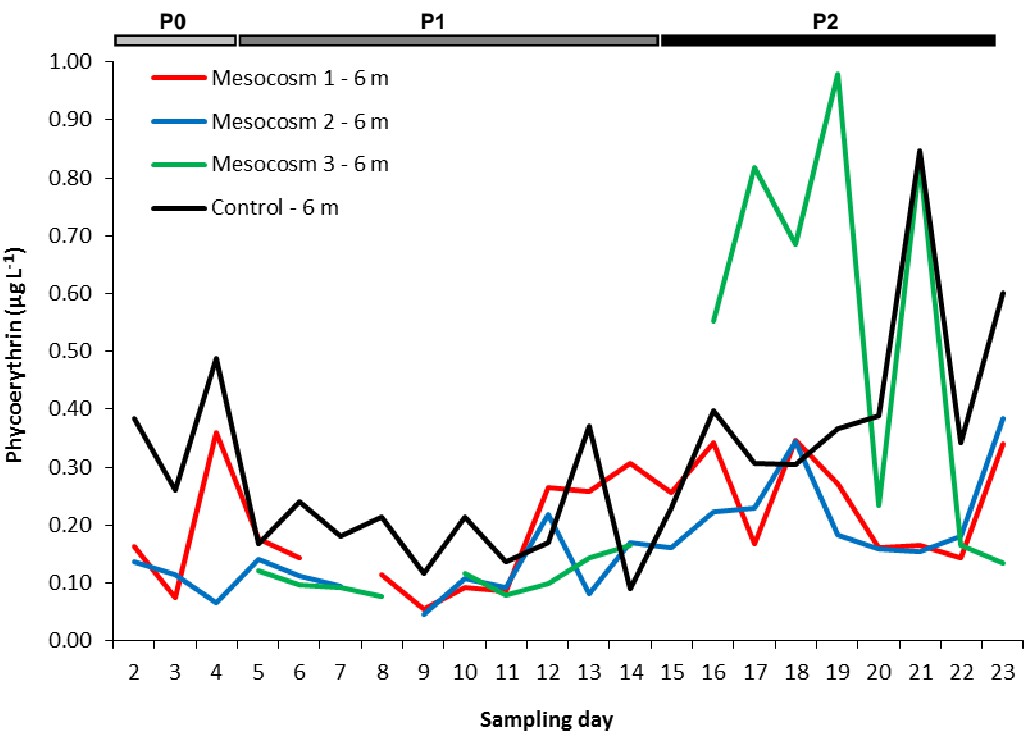

Figure 3 : Phycoerythrin in µg L$^{-1}$ at the intermediate depth (6 m) inside each mesocosm and in the control area outside of mesocosms.

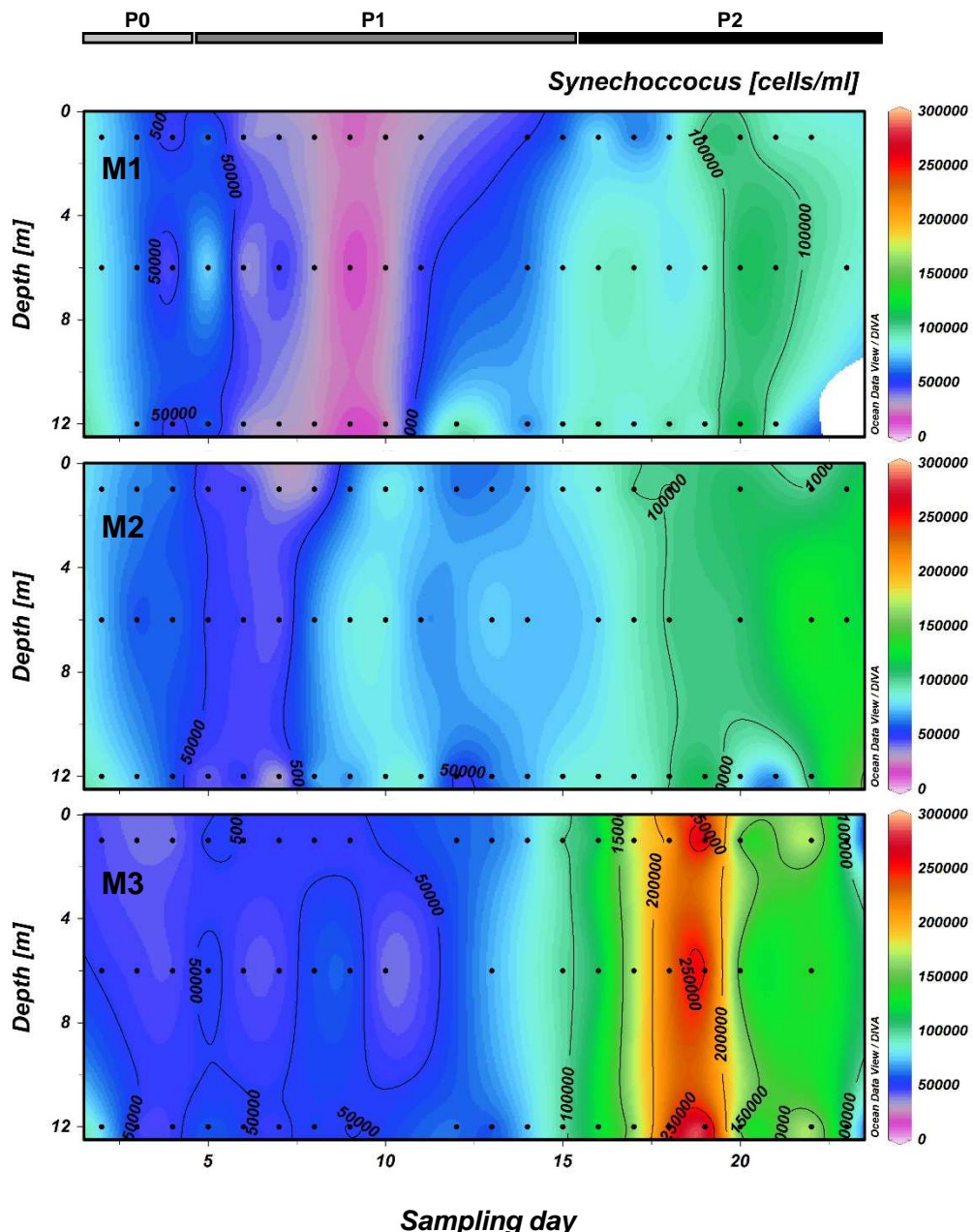

Figure 4 : *Synechococcus* in cells mL$^{-1}$ at each of the three depths (1, 6 and 12m) inside each mesocosm (M1, M2 and M3).

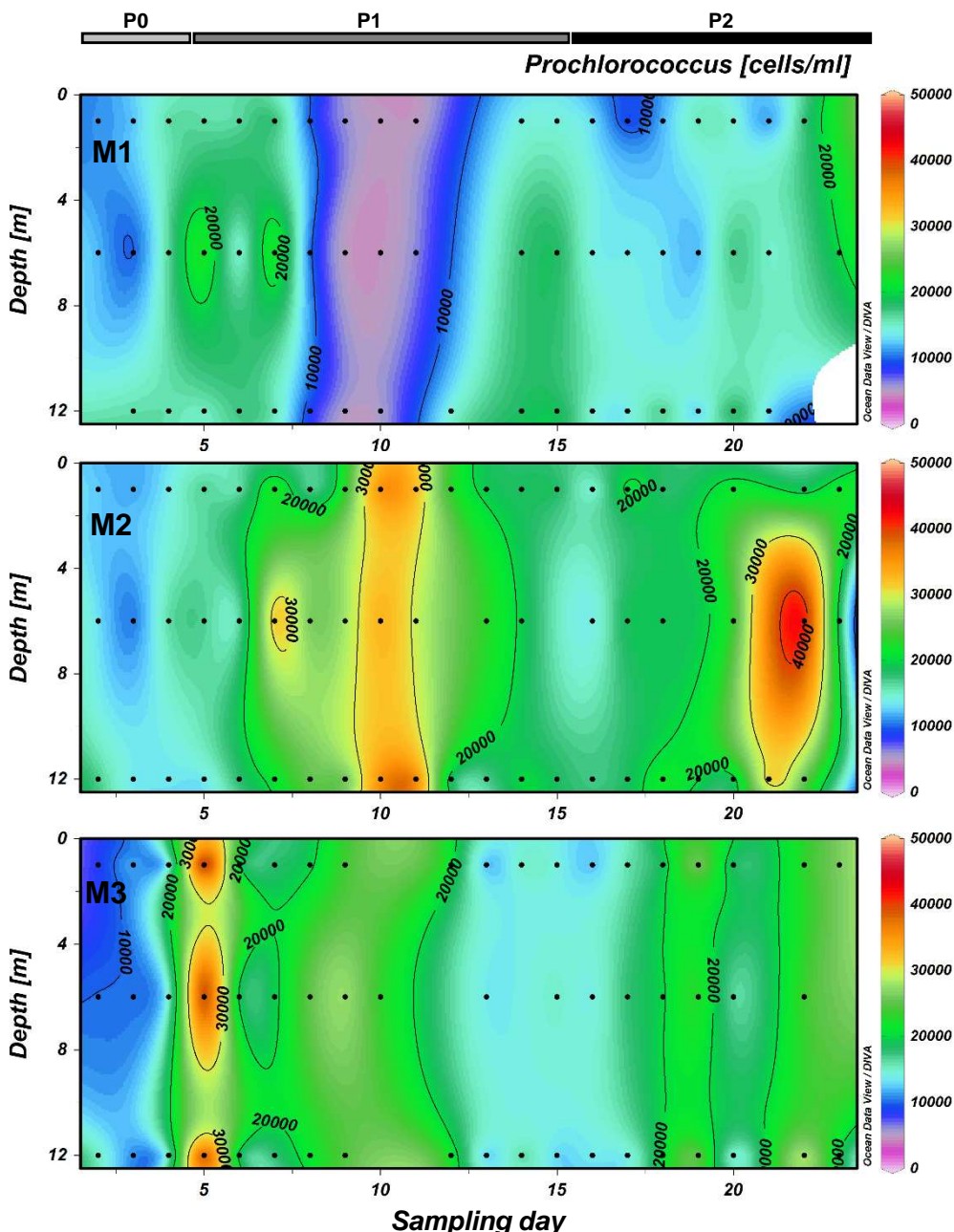

Figure 5 : *Prochlorococcus* in cells mL$^{-1}$ at each of the three depths (1, 6 and 12m) inside each mesocosm (M1, M2 and M3).

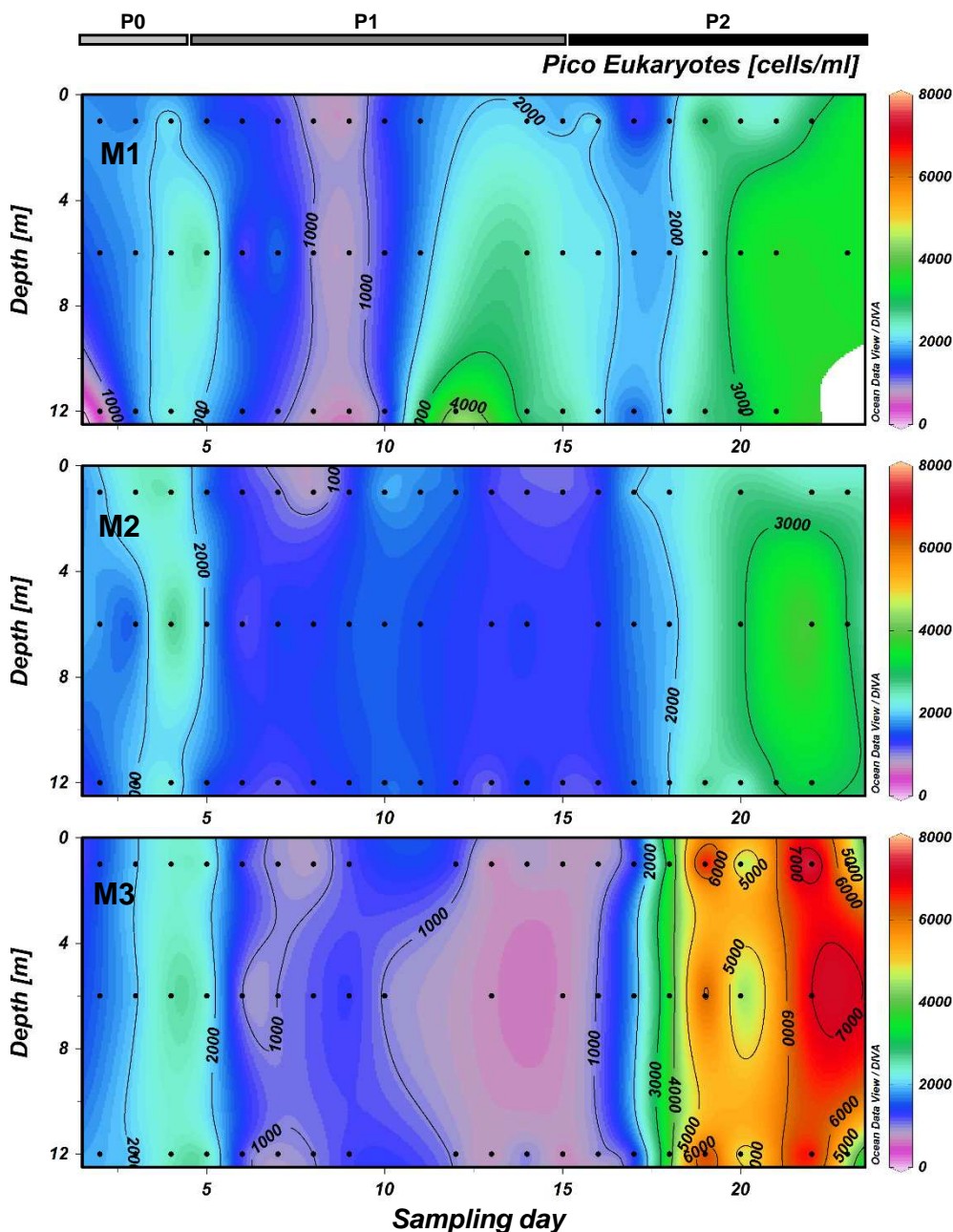

Figure 6 : Pico-phytoeukaryotes in cells mL$^{-1}$ at each of the three depths (1, 6 and 12m) inside each mesocosm (M1, M2 and M3).

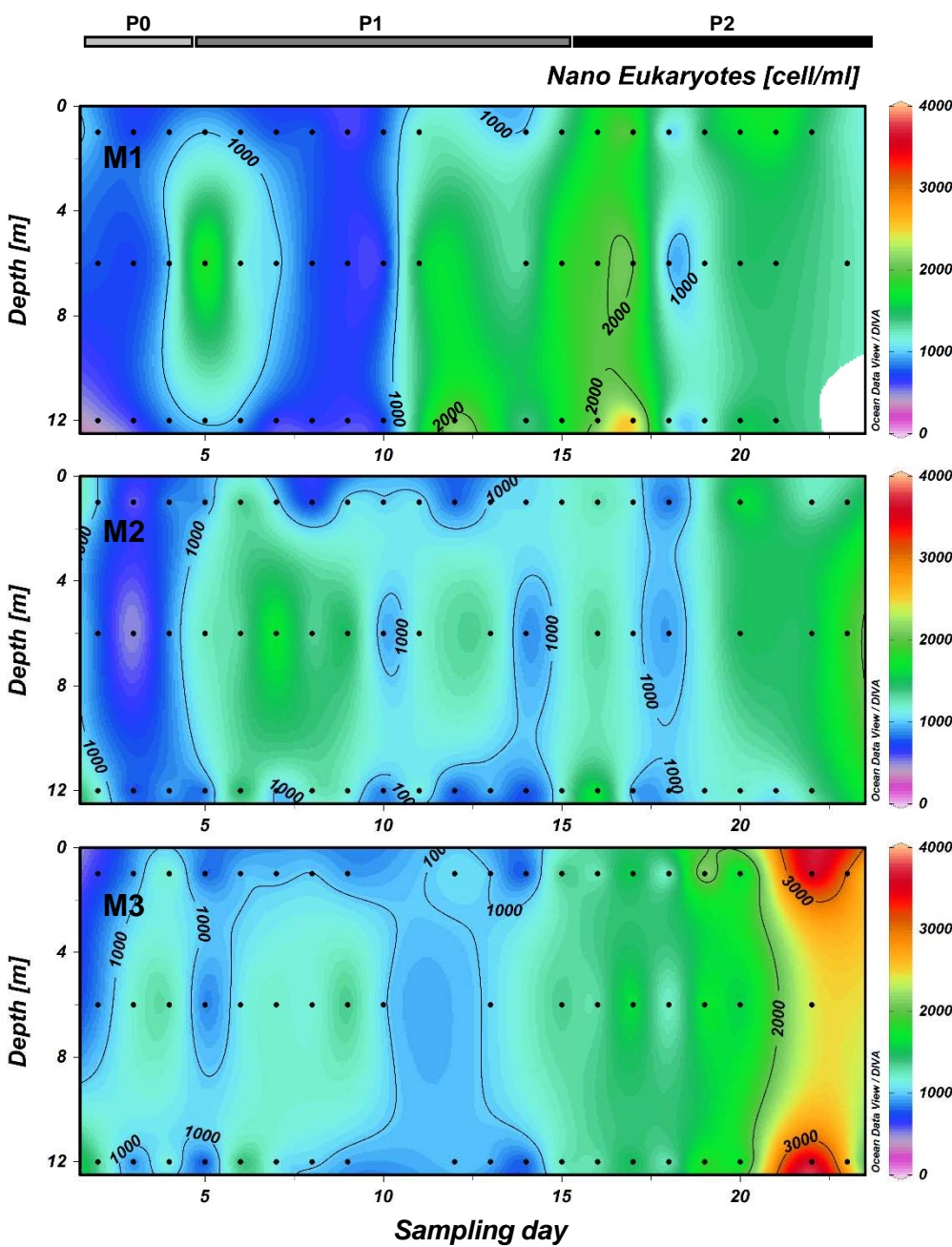

Figure 7 : Nano-phytoeukaryotes in cells mL$^{-1}$ at each of the three depths (1, 6 and 12m) inside each mesocosm (M1, M2 and M3).

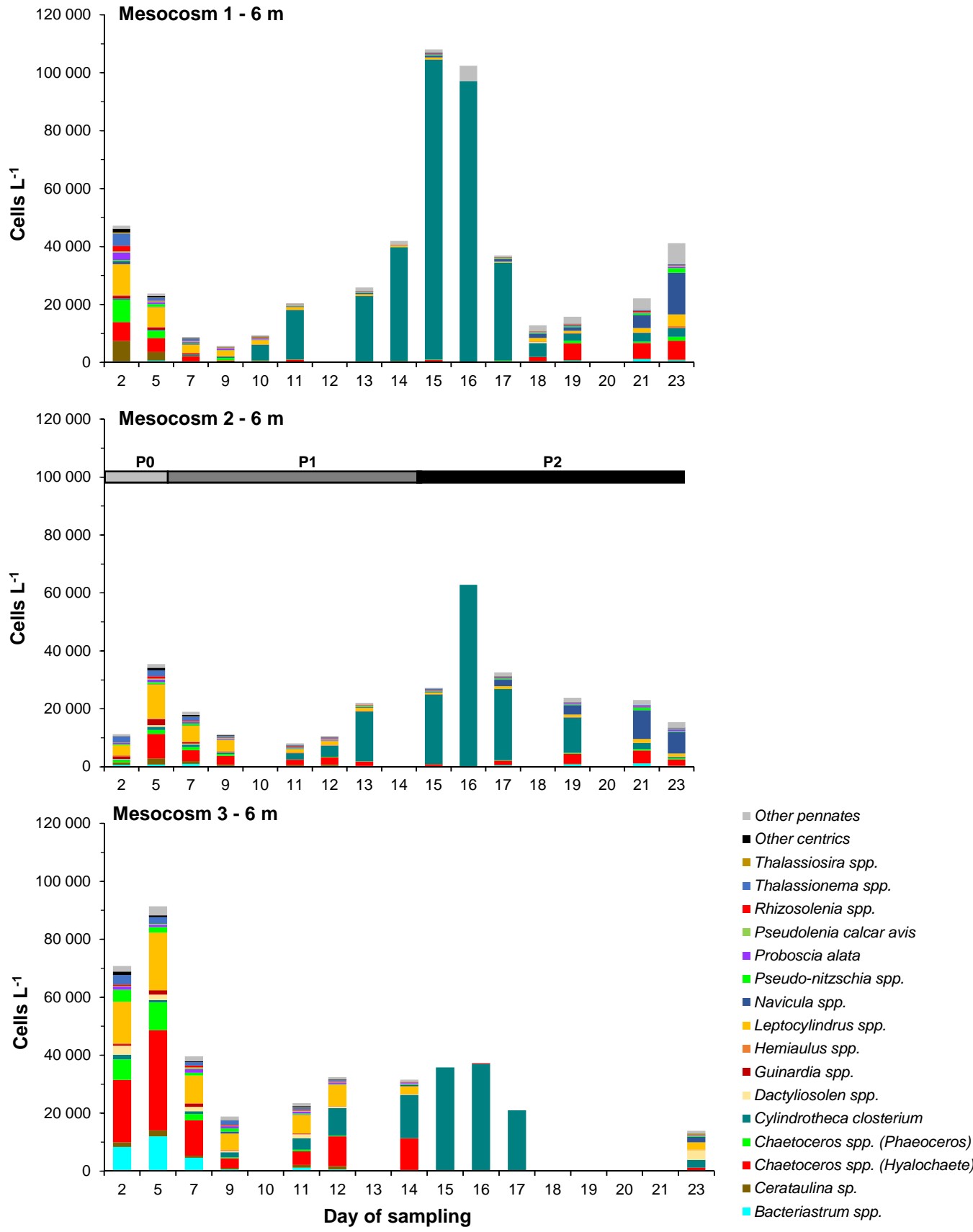

Figure 8 : Diatom genera/species abundance in cells L$^{-1}$ at the intermediate depth (6 m) in each mesocosm.

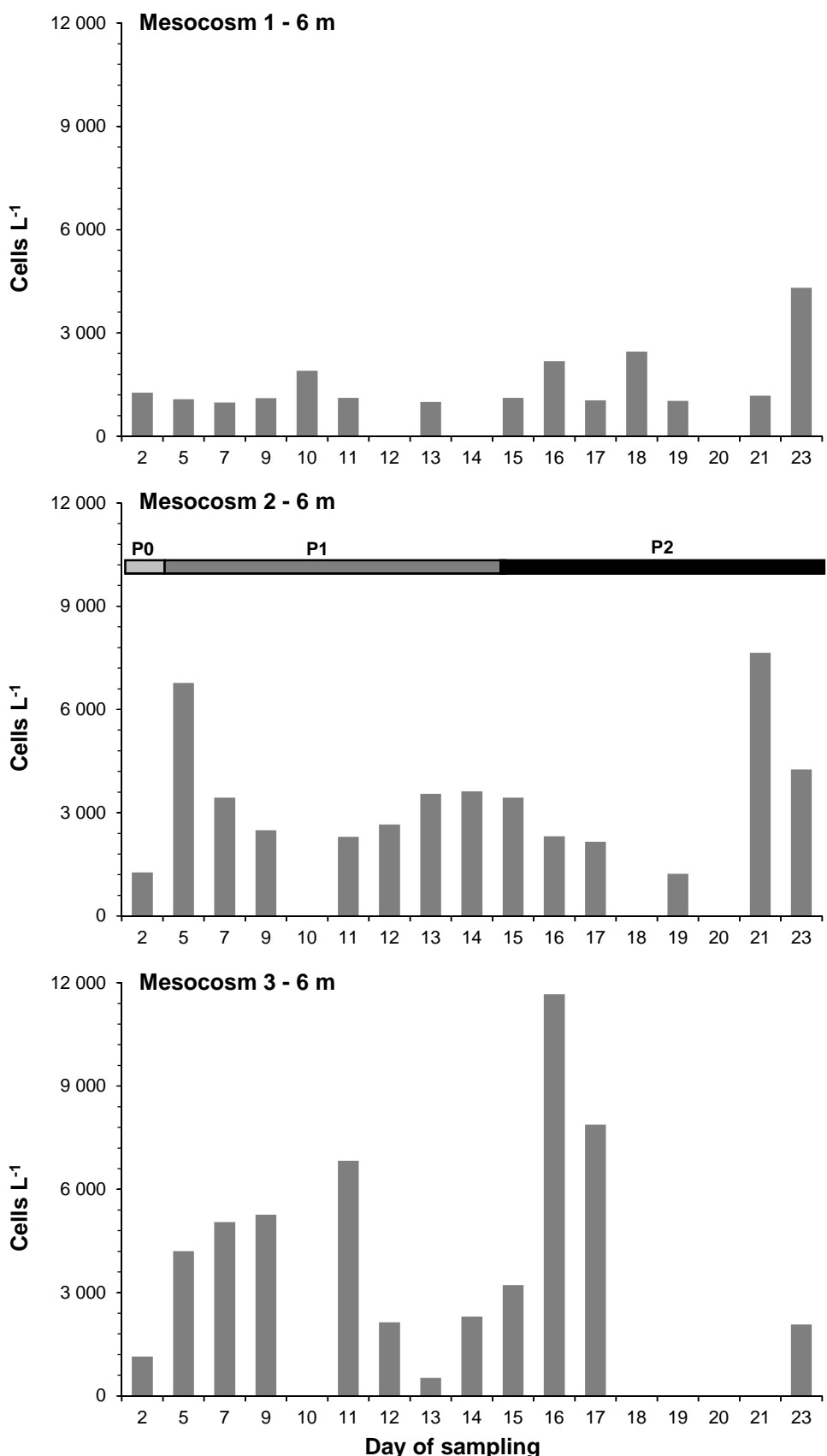

Figure 9 : Total dinoflagellate abundance (in cells L$^{-1}$) at the intermediate depth (6 m) inside each mesocosm.

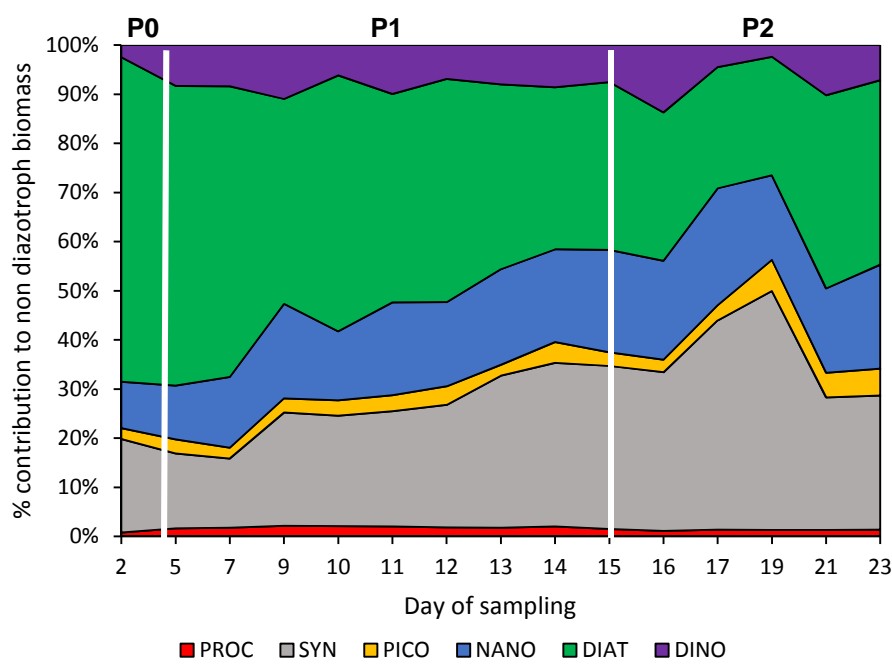

Figure 10 : Dynamics of the biomass of the main groups constituting phytoplankton communities (diazotrophs not included) over the course of the experiment for *Prochlorococcus* (PROC), *Synechococcus* (SYN), pico-phytoeukaryotes (PICO), nano-phytoeukaryotes (NANO), diatoms (DIAT) and dinoflagellates (DINO).

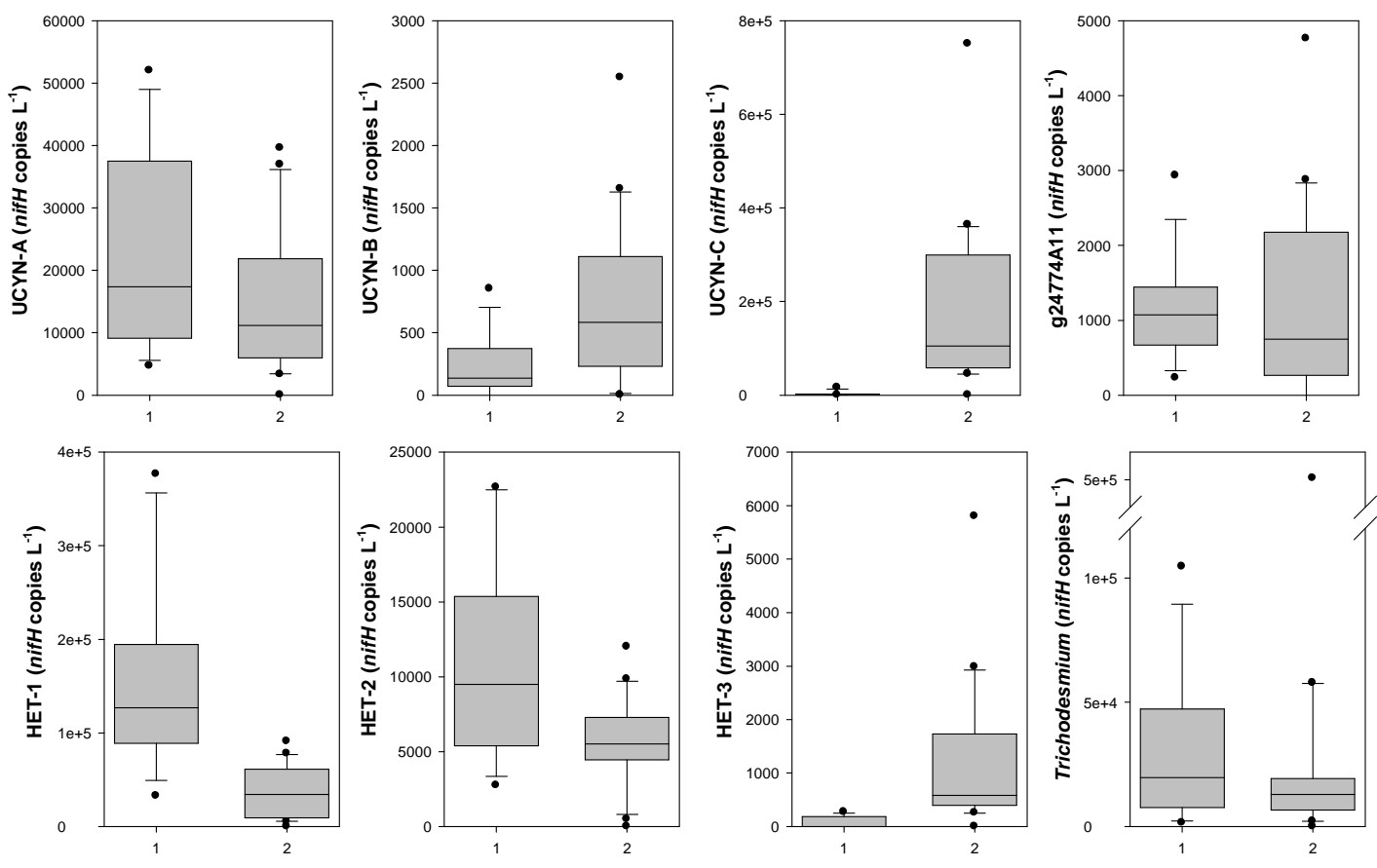

Figure 11 : Boxplots of the main diazotrophs groups in *nifH* gene copies L$^{-1}$ in the three mesocosms during the two periods P1 and P2.

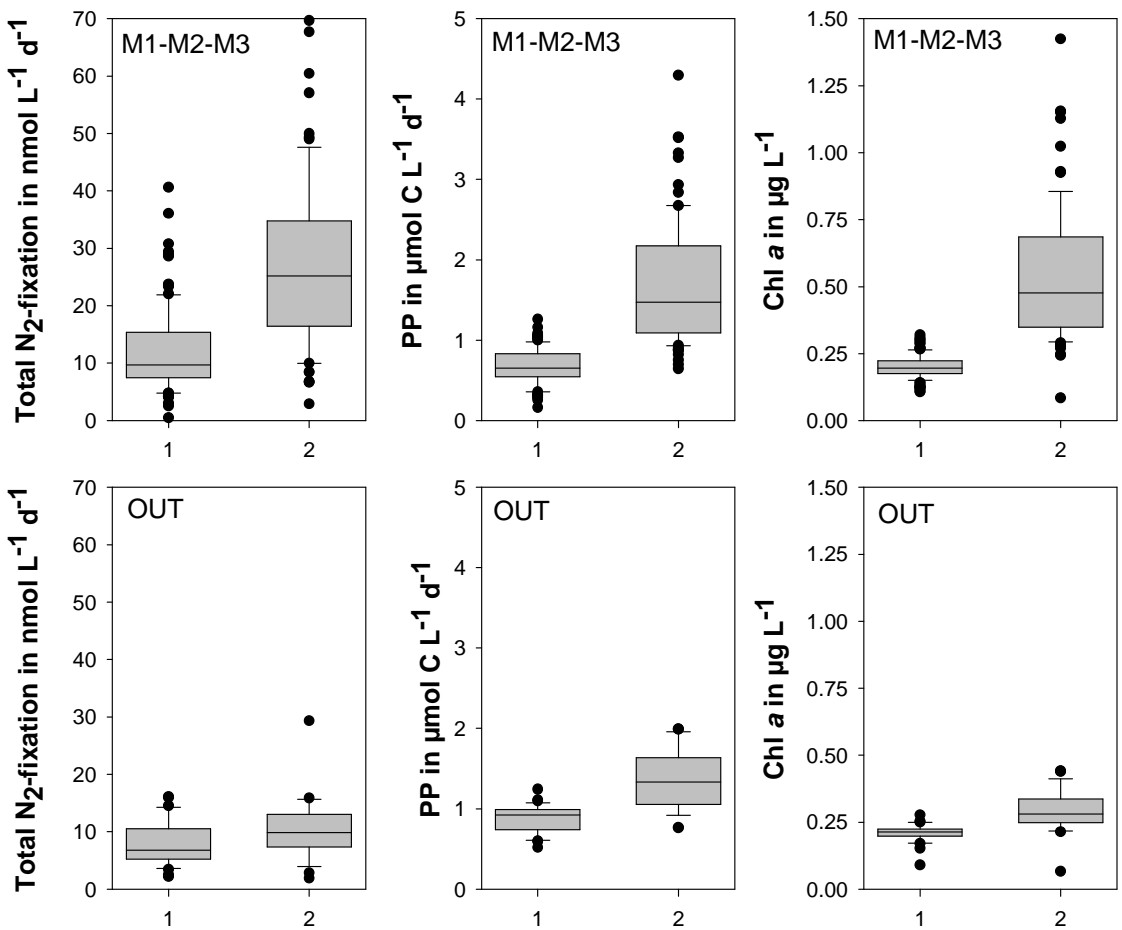

Figure 12 : Boxplots of primary production (in µmol C L$^{-1}$ d$^{-1}$), nitrogen-fixation rates (in nmol N L$^{-1}$ d$^{-1}$) and in Chl *a* concentrations (in µg L$^{-1}$) in the three mesocosms (top pannels) and in the lagoon waters outside of mesocosms (bottom pannels) during the two periods P1 and P2.

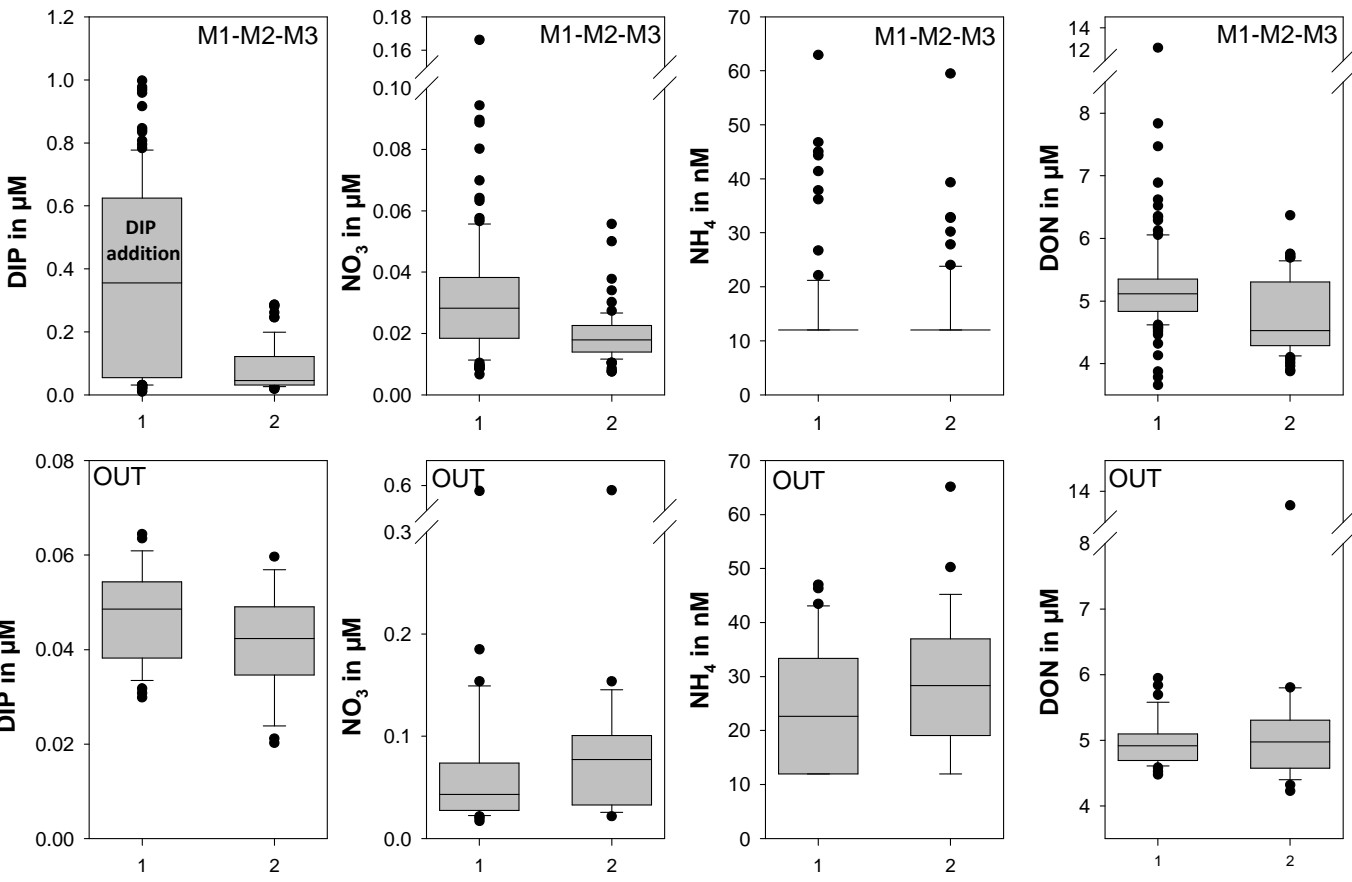

Figure 13 : Boxplots of nutrients, DIP, NO$_3$, DON in µM and NH$_4^+$ (in nM) in the three mesocosms (top pannels) and in the lagoon waters outside of mesocosms (bottom pannels) during the two periods P1 and P2.

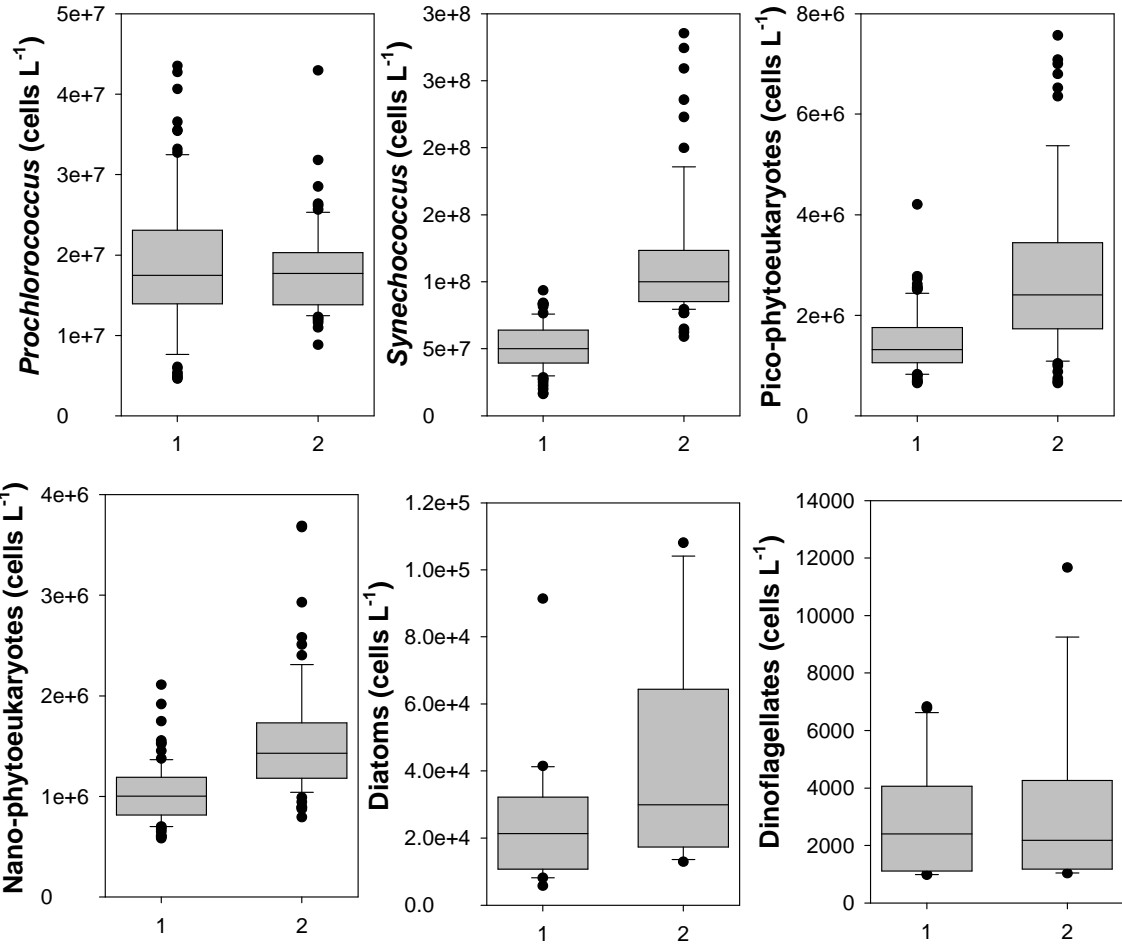

Figure 14 : Boxplots of the main phytoplanktonic groups in cells L$^{-1}$ during the two periods P1 and P2.

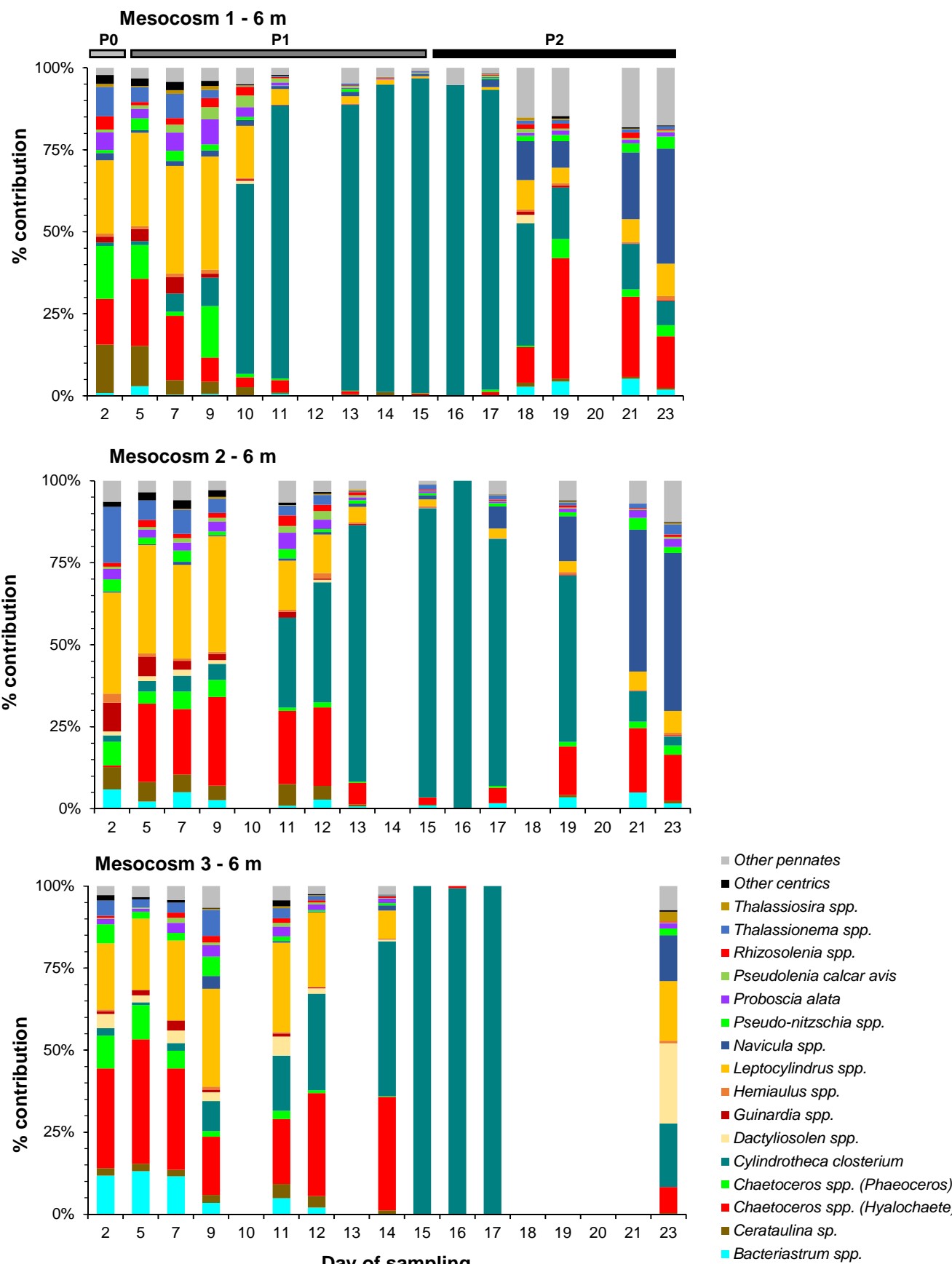

Figure S1: Main diatom genera/species composition in % contribution at the intermediate depth (6 m) in each mesocosm.

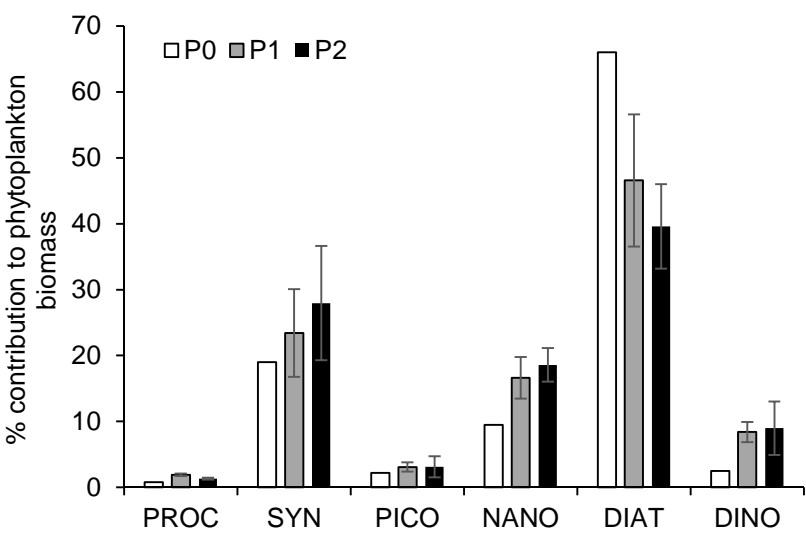

Figure S2 : Average (±SD) contribution to C biomass of the main groups constituting phytoplankton communities (diazotrophs not included) over the course of the experiment following the three periods P0, P1 and P2 for *Prochlorococcus* (PROC), *Synechococcus* (SYN), pico-phytoeukaryotes (PICO), nano-phytoeukaryotes (NANO), diatoms (DIAT) and dinoflagellates (DINO).