# Peer review of "Phytoplankton community structure in the VAHINE mesocosm experiment"

_Biogeosciences, 2015_

## Referee Comment (RC1) · Anonymous Referee #1 · 23 Feb 2016

Leblanc and colleagues present floristic results from a LNLC mesocosm experiment in Noumea designed to stimulate diazotrophy and follow the transfer of newly fixed N through the ecosystem. Specifically, they present data on chlorophyll a and phyco-erythrin pigments and the abundances of pico- and nano-phytoplankton, diatoms and dinoflagellates. Following a lag period, the DIP-treated mesocosms responded with increased pigments overall, and a notable increase in Synechococcus and a decrease in diatoms. Diatom species composition was also affected.

The pigment and phytoplankton data in this manuscript represent a tremendous amount of careful work and should be published in some form. However, I am having difficulty reviewing this as a standalone work. It reads more like a collection of results or a data report than a cohesive paper. Because this manuscript appears to be part of a special volume, it may be that the importance of these measurements in the

overall context of the mesocosm study would become clearer when the whole volume is considered. However, I cannot review it as such. The separation of material into individual papers appears to have been done in a rather awkward fashion. For example, this paper on phytoplankton carefully excludes diazotroph abundances – why? The whole point was to stimulate diazotrophs, and they may have become an important part of the phytoplankton community – indeed, the phycoerythrin results suggest so. In addition, data from other papers are inserted here without explanation: e.g., measurements of N2 fixation rates, nutrient concentrations and nifH gene copy numbers in Figs. 11, 12 and S3 that were never described in the Methods section. Also, much of the Discussion section focuses on explaining results that do not appear in the present manuscript.

Technically, the paper is clearly written and the figures are nicely constructed. Although, I do not see the value of the contour plots (Figs. 2, 4-7), especially since there do not appear to be any clear depth-dependent patterns that I can see nor any discussion of depth effects on any of the measured parameters. Each 4-panel contour figure could be presented more effectively as a line plot like Fig. 3 with depth-averaged values, or, alternatively, as a box plots as in Fig. 13.

In short, I do not think this is a complete manuscript on its own, especially if diazotroph abundances are not included and other data sets are pulled from other manuscripts without explanation. I recommend that the authors reconsider how they divide up the experimental results between manuscripts. The data presented here may be best included within a more cohesive work.

---

## Referee Comment (RC2) · Anonymous Referee #2 · 14 Apr 2016

This manuscript provides results of the changes in phytoplankton community composition during a mesocosm experiment in Low Nutrient, Low Chlorophyll waters in the Southwest Pacific. The primary objective was to stimulate a bloom of N2-fixing cyanobacteria through PO$ addition and track the resulting particulate carbon fluxes resulting either directly for the N2-fixing organisms (including those associated with diatoms) or indirectly through the diazotrophs providing a source of fixed, reduced nitrogen to the enclosed system. The manuscript by Leblanc et al. provides a thorough overview of the changes in non-diazotroph phytoplankton communities within three replicate mesocosms during three phases of development. Phase one of the analysis follows a spike of PO4 to each mesocosm that was meant to stimulate N2-fixation. The second phase corresponded with a transition from N2-fixing cyanobacteria to non-N2-fixers, following the depletion of the PO4 spike. From the data presented, there ap-

pears to by some rather complex dynamics of the phytoplankton communities not only throughout the duration of the experiment but among the mesocosms. For example, it is apparent that mesocosm 3 achieved higher amounts of phytoplankton biomass during phase 2 of the experiment associated with increases in pico- and nano-eukaryotes that were not replicated in the other experiments. In addition, the substantial increase in phycoerythrin in mesocosm 3 that was somewhat replicated in the control sampling but not found in the other incubations is somewhat perplexing.

I found the article well-written and for the most part, the methods used seem applicable for the general objectives of the paper in detailing the phytoplankton taxonomy transitions throughout the bloom. The manuscript is more of a descriptive account of the phytoplankton successions rather than being able to provide definitive reasoning as to why certain groups of phytoplankton changed in abundance when they did. Noticeably absent are the changes in abundance in N2-fixers at the same resolution as what is presented for the non-diazotroph phytoplankton groups apart from phase averages in N2-fixation rates in figure 11 and gene expression data in the supplemental. Although I gather this is due to another paper that will be part of the same special issue describing these results (Turk-Kubo et al. 2015). In addition, it is also surprising that chlorophyll concentrations do not necessarily provide much insight into the phytoplankton composition dynamics. In fact, during the peak abundance of the C. closterium around days 15 and 16, chlorophyll concentrations remained quite low. Chlorophyll concentrations seem to be better correlated with pico and nano-eukaryotes that bloomed near the end of phase 2 of the observation period. In addition, the conversion to carbon biomass presented in Figure 10 was rather unsatisfactory. It would have been interesting to compare these biomass estimates with those that were exported to determine whether similar proportions exist between the two or if there is a preferential export of certain groups (e.g. diatoms etc.). Thus in some regards, I find the results presented to be somewhat incomplete. However, as part of a special issue dedicated to the VAHINE mesocosm experiment, this manuscript does constitute an important contribution to describing the overall outcomes. Although further support and discussion on principle

factors governing the transitions observed in phytoplankton group successions would benefit the manuscript immensely.

Specific Comments:

Line 62 – Given the primary outcome was to determine whether a diazotroph bloom would increase C export fluxes to depth, I am curious to know if this was the case. Although I can imagine this will be presented in other papers (possibly lead by Bonnet?), perhaps a short discussion as to whether this was the case would add value to this manuscript. Clearly, the resulting changes in non-diazotroph abundances would have also contributed to the overall outcome in influencing C export potential.

Line 79 – So do the authors think that the transition from DDAs to cyanobacterial N2-fixers may have influenced the phytoplankton composition somehow?

Line 161 – Include the volumes settled for micro-phytoplankton enumeration.

Line 170 – It is unfortunate that cell volume estimates of diatoms counted were not measured as this can vary substantially for a given species. It should therefore be noted that these C biomass estimates be taken with extreme caution.

Line 218 – . . .were comprised of between. . .

Line 347 – It is surprising there is little discussion of Si limitation of diatom growth. Clearly with Si concentrations below 2 uM, this would favor very lightly silicified species (such as C. closterium) or non-diatom phytoplankton groups. I would guess that after PO4 addition, Si is a major regulator of diatom growth (as well as possibly Fe).

Line 400 – Is there evidence to support this hypothesis within the scientific literature? Why would C. closterium have such a higher NH4 affinity. It's likely more related to their low Si requirements relative to other diatoms.

Line 409 – It's also very likely that these dinoflagelates were mixotrophic. Gyrodinium/Gymnodinium are well known to exhibit heterotrophy within low nutrient environments when this mode is more favorable.

Line 463 – Indeed, this might be the case for a number of phytoplankton groups and not just Synechococcus.

Line 515 – remove "to " in benefited to the entire. . .

Line 517 – observed by, not on.

Line 518 – More precisely, what are these clear implications for the efficiency of C export by DDN? What groups were exported?

Figure 5 – Missing M labels of panels to be consistent with other figures.

---

## Referee Comment (RC3) · Anonymous Referee #3 · 17 Apr 2016

This papers describes part of a very interesting and exhausitve study about phytoplankton succession and phenology during a mesocosm experiment in LNLC waters. The presented data on floristic identification and pigment analysis is extremely well presented and documented. However, being part of a more exhaustive study containing important complementary data (nutrients, biological processes as production, molecular biology, other diazotroph identifications, TEP concentrations, sediment traps...), it seems very difficult to have an good view of the overall work and results from this study. For instance, there is a clear gap between the "results" section (phytoplankton and pigment results) and the "discussion" section where all other data is combined but not always presented. Most of these other results are described in parallel papers.

It seems to me that this work clearly desserves to be published. However, results shoud be re-structured differently in order to give a broad and complete view and understanding of phytoplankton /ecosystem functionning. I therefore suggest the autors to review the structure of the paper by presenting the required database and information on complementary and eseential measured data.

———————————————————

---

## Author Comment (AC1)

**Please find in blue text below our response to all three reviewers.**

**Anonymous Referee #1**

Leblanc and colleagues present floristic results from a LNLC mesocosm experiment in Noumea designed to stimulate diazotrophy and follow the transfer of newly fixed N through the ecosystem. Specifically, they present data on chlorophyll a and phycoerythrin pigments and the abundances of pico- and nano-phytoplankton, diatoms and dinoflagellates. Following a lag period, the DIP-treated mesocosms responded with increased pigments overall, and a notable increase in Synechococcus and a decrease in diatoms. Diatom species composition was also affected. The pigment and phytoplankton data in this manuscript represent a tremendous amount of careful work and should be published in some form. However, I am having difficulty reviewing this as a standalone work. It reads more like a collection of results or a data report than a cohesive paper. Because this manuscript appears to be part of a special volume, it may be that the importance of these measurements in the overall context of the mesocosm study would become clearer when the whole volume is considered. However, I cannot review it as such. The separation of material into individual papers appears to have been done in a rather awkward fashion. For example, this paper on phytoplankton carefully excludes diazotroph abundances – why? The whole point was to stimulate diazotrophs, and they may have become an important part of the phytoplankton community – indeed, the phycoerythrin results suggest so. In addition, data from other papers are inserted here without explanation: e.g., measurements of N2 fixation rates, nutrient concentrations and nifH gene copy numbers in Figs. 11, 12 and S3 that were never described in the Methods section. Also, much of the Discussion section focuses on explaining results that do not appear in the present manuscript.

We understand the reviewer's point of view about the separation of results in this special issue, and this paper is likely suffering from the intent not to repeat too many results presented in the other papers. The separation between results for this special issue was a fruit of long discussions between potential co-authors, and the focus of the experiment being the study of diazotrophs and the fate of derived DDN, several other papers already presented the in lengthy details the diazotroph community, using very different techniques based on qPCR (Turk-Kubo et al.), 16S tag sequencing (Pfreundt et al.,a) and metatranscriptomic to investigate the microbial gene expression dynamics from diazotrophic and non-diazotrophic taxa (Pfreundt et al.,b). These techniques need to be described at some length and we felt that the inclusion of all other taxonomic and pigment information there would have resulted in too large papers that would have lost focus. This is the reason we felt that all other taxonomic data except for diazotrophs (that we yet chose to show in the supplementary material as average boxplots for the three periods in order not to show results similarly to these other papers) could warrant for another "community structure" paper that would complement the main information provided in Turk-Kubo et al. and Pfreundt et al. If the reviewer feels that these data can not stand alone despite the fact that they are clearly included in a special issue and complementary to other papers, we propose to rewrite part of the paper as to include the diazotrophs group as a whole, maybe with some distinction between "total diazotrophs","total UCYN" and "total filamentous" as the fine description of the succession within each group is already described fully in Turk-Kubo et al. (See also my response to reviewer #3 on the last page answering this same question).

Technically, the paper is clearly written and the figures are nicely constructed. Although, I do not see the value of the contour plots (Figs. 2, 4-7), especially since there do not appear to be any clear depth-dependent patterns that I can see nor any discussion of depth effects on any of the

measured parameters. Each 4-panel contour figure could be presented more effectively as a line plot like Fig. 3 with depth-averaged values, or, alternatively, as a box plots as in Fig. 13.

We agree that the ODV contour plots could be easily replaced by line plots, we chose this graphic output because many other ODV plots were presented in the companion papers and felt that it provided some homogeneity in the special issue, but line plots can easily be redrawn if needed.

In short, I do not think this is a complete manuscript on its own, especially if diazotroph abundances are not included and other data sets are pulled from other manuscripts without explanation. I recommend that the authors reconsider how they divide up the experimental results between manuscripts. The data presented here may be best included within a more cohesive work.

We agree to include diazotroph data, and also better describe in the methods section the different biogeochemical data used. Again, these are described in full in the companion papers (Berthelot et al., Bonnet et al.a, b) and the aim of the paper was not to rescribe the entire biogeochemical environments in the results, but rather use the main fluxes average to explain / caracterize the different planktonic succession phase. If needed, we can add sections to the material and methods to describe how nutrient stocks, primary production and $N_2$ fixation fluxes were obtained for instance.

**Anonymous Referee #2**

This manuscript provides results of the changes in phytoplankton community composition during a mesocosm experiment in Low Nutrient, Low Chlorophyll waters in the Southwest Pacific. The primary objective was to stimulate a bloom of N2-fixing cyanobacteria through PO4 addition and track the resulting particulate carbon fluxes resulting either directly for the N2-fixing organisms (including those associated with diatoms) or indirectly through the diazotrophs providing a source of fixed, reduced nitrogen to the enclosed system. The manuscript by Leblanc et al. provides a thorough overview of the changes in non-diazotroph phytoplankton communities within three replicate mesocosms during three phases of development. Phase one of the analysis follows a spike of PO4 to each mesocosm that was meant to stimulate N2-fixation. The second phase corresponded with a transition from N2-fixing cyanobacteria to non-N2-fixers, following the depletion of the PO4 spike. From the data presented, there appears to by some rather complex dynamics of the phytoplankton communities not only throughout the duration of the experiment but among the mesocosms. For example, it is apparent that mesocosm 3 achieved higher amounts of phytoplankton biomass during phase 2 of the experiment associated with increases in pico- and nano-eukaryotes that were not replicated in the other experiments. In addition, the substantial increase in phycoerythrin in mesocosm 3 that was somewhat replicated in the control sampling but not found in the other incubations is somewhat perplexing.

It is true that there is some degree of variability between mesocosms, which can hardly be avoided. Replicability for such large volume experiments and over >3 weeks time is always difficult to obtain. From the literature, slight divergence in biological and chemical evolution among different replicated mesocosms is not uncommon, particularly after the first week of enclosure (Martinez-Martinez et al., 2006; Pulido-Villena et al., 2014). However we feel that this variability is not so important that it undermines the main results of the VAHINE experiment, which successfully triggered a diazotroph bloom, and allowed to follow the fate of DDN through the food web and in

the downward export flux. A paragraph (section 2.5, together with Table 1) in Bonnet et al's introductory paper deals with mesocosms variability. We reproduce some of it below, and argue that Bonnet et al successfully make the case that the degree of variability observed is acceptable and that globally the three mesocosms were well reproduced in their main patterns.

"For example, bulk $N_2$ fixation rates averaged 18.5$\pm$1.1 nmol NL$_{-1}$ d$_{-1}$ (standard deviation was calculated on the average $N_2$ fixation rates of each mesocosm) over the 23 days of the experiment (all depths averaged together). $N_2$ fixation rates did not differ significantly among the three mesocosms ($p<0:05$; Kruskall–Wallis test; Berthelot et al., 2015). Moreover, we consistently observed the same temporal dynamics over the three mesocosms, such as the dramatic increase of rates from days 15 to 23 (during which they reached 27.3$\pm$1.0 nmol N L$_{-1}$ d$_{-1}$. This together indicates good replicability between the mesocosms (Bonnet et al., 2015). Molecular data also report a shift in the diazotrophic community composition around day 15, with a bloom of UCYN-C consistently occurring in the three mesocosms (see Turk-Kubo et al., 2015). The same feature was observed for *Synechococcus* abundances, which increased by a factor of 2 from day 15 to day 23 in every mesocosm (Leblanc et al., 2016). Finally, the diatom community, which was very diverse during the first half of the experiment, suddenly shifted beginning at day 10, and *Cylindrotheca closterium* consistently became the dominant diatoms in the three mesocosms (Leblanc et al., 2016). These observations, together with the CV reported above, indicate that biogeochemical and biological conditions were comparable between the three mesocosms."

In addition to this, the initial conditions prevailing before the DIP enrichment could be also at the origin of slight divergence. Indeed mesocosms were closed 3 days before the DIP addition, and many species of diazotrophs exhibit a patchy distribution (Bombar et al., 2015). Hunt et al. (2016) also noticed larger amounts of zooplankton individuals in M3 at the beginning of the experiment, some of which, stressed by the mesocosms, might have died (some larger amounts of 'swimmers' were recovered in the traps in M3), contributing to supplementary sources of N in M3. This might explain why M3 was more different than the other 2 mesocosms.

I found the article well-written and for the most part, the methods used seem applicable for the general objectives of the paper in detailing the phytoplankton taxonomy transitions throughout the bloom. The manuscript is more of a descriptive account of the phytoplankton successions rather than being able to provide definitive reasoning as to why certain groups of phytoplankton changed in abundance when they did. Noticeably absent are the changes in abundance in N2-fixers at the same resolution as what is presented for the non-diazotroph phytoplankton groups apart from phase averages in N2-fixation rates in figure 11 and gene expression data in the supplemental. Although I gather this is due to another paper that will be part of the same special issue describing these results (Turk-Kubo et al. 2015).

Indeed diazotroph succession data is already presented in two other papers (Turk-Kubo et al. this 2015) and (Pfreundt et al, this issue). Kendra Turk-Kubo agrees to add these data again here in the main result section. We can argue that they would be presented here in a very synthetic way (boxplot figures of P1 and P2 – the figure in supplement would be promoted up to results and discussed more in link with the other data to get a sense of the entire phytoplanktonic community. Some brief description of methods could be added too.

In addition, it is also surprising that chlorophyll concentrations do not necessarily provide much insight into the phytoplankton composition dynamics. In fact, during the peak abundance of the C. closterium around days 15 and 16, chlorophyll concentrations remained quite low. Chlorophyll concentrations seem to be better correlated with pico and nano-eukaryotes that bloomed near the end of phase 2 of the observation period. In addition, the conversion to carbon biomass presented

in Figure 10 was rather unsatisfactory. It would have been interesting to compare these biomass estimates with those that were exported to determine whether similar proportions exist between the two or if there is a preferential export of certain groups (e.g. diatoms etc.).

We agree that biomass estimates for the entire community is always a difficult exercise. But we feel that Figure 10 allowed to quickly assess in very visual the actual contribution of each group, which is biased by the very wide abundance ranges presented in the previous figures. The relative contributions to PROC and SYN for instance are much clearer here, with PROC contributing very little to biomass, where as they are the second most abundant organism, of course due to their small sizes.

As explained below to your specific comment below, this paper was not dedicated to comparison of the community succession and impact on export, as it is at length described in Bonnet et al.this issue and Knapp et al.this issue Also it is not possible to disentangle from trap POC which group contributes more to biomass, except for $N_2$-fixers which was done using molecular biology techniques. See below for more detail on POC export calculations.

Thus in some regards, I find the results presented to be somewhat incomplete. However, as part of a special issue dedicated to the VAHINE mesocosm experiment, this manuscript does constitute an important contribution to describing the overall outcomes. Although further support and discussion on principle factors governing the transitions observed in phytoplankton group successions would benefit the manuscript immensely.

***Specific Comments:***

Line 62 – Given the primary outcome was to determine whether a diazotroph bloom would increase C export fluxes to depth, I am curious to know if this was the case. Although I can imagine this will be presented in other papers (possibly lead by Bonnet?), perhaps a short discussion as to whether this was the case would add value to this manuscript. Clearly, the resulting changes in non-diazotroph abundances would have also contributed to the overall outcome in influencing C export potential.

Export of diazotroph and non diazotroph is described in Bonnet et al. (see Fig 4 below) as well as the export efficiency for each group, and also discussed in Knapp et al (this issue). Some insertion in the discussion can be added, but again we might get comments back that POC export measurements need to be presented and detailed in the method / results section, but is is already presented in two other papers in the special issue. We feel again that this paper was not meant to be a synthesis paper of the VAHINE experiment, already successfully published in this issue by Bonnet et al in both an introductory and result papers, and also in Berthelot et al., but merely a description of the rest of the community along side the very focused papers on $N_2$ fixation fluxes and diazotrophic communities. See below for what is already described in Bonnet's paper, maybe a short reference to these results in the discussion would suffice. The calculations say that according to qPCR quantification of diazotrophs in the sediment traps and in the water column, ~10 % of UCYN-C from the water column was exported to the traps daily. Based on microscopic observations, UCYN-C abundances counted in traps material was converted in carbon biomass based on cell to carbon conversion factors and represented as much as 22.4 ± 5.5 % of the total POC exported at the height of the UCYN-C bloom. So a 10 % average export of UCYN-C relative to POC to the traps and a maximum value of 22 % and up to 7 % for the DDAs. The similar calculations can not be made for other non $N_2$-fixing groups, as they were not enumerated in the traps (in fecal pellets, aggregates etc…).

[Figure]

(c)

**Figure 4. (a)** Abundance of UCYN-C (nifH copies $L^{-1}$) and **(b)** other nifH phylotypes (UCYN-A2, UCYN-B, Trichodesmium, het-1, het-3) (nifH copies $L^{-1}$) recovered in the sediment trap on day 17 and 19. **(c)** Proportion of POC export associated with diazotrophs in the sediment traps on day 17 in M2 (height of UCYN-C bloom).

Figure from Bonnet et al. this issue : "Thus, our data emphasize that, despite their small size relative to DDAs, UCYN-C are able to directly export organic matter to depth by forming densely populated aggregates that can rapidly sink. This observation is further confirmed by the e ratio, which quantifies the efficiency of a system to export POC relative to primary production (e ratio=POC export/PP) and was significantly higher ($p < 0.05$) during P2 (i.e., during the UCYN-C bloom; 39.7 ± 24.9 %) than during P1 (i.e., when DDAs dominated the diazotrophic community; 23.9 ± 20.2 %) (Berthelot et al., 2015b)."

Line 79 – So do the authors think that the transition from DDAs to cyanobacterial N2-fixers may have influenced the phytoplankton composition somehow?
It is very difficult to nail which factor influenced each group succession as we believe is a result of a complex interplay between abiotic factors and biological interactions between groups. Clearly though, we think DDAs receded because the DIP addition allowed for other groups to dominate and for unicellular diazotroph to increase after a certain lag time, but looking at the non diazotroph biomass plot (fig 10), only the SYN and diatom biomass showed significant evolution over time.

Line 161 – Include the volumes settled for micro-phytoplankton enumeration.
The following line will be added: "Sedimented volume was on average 140 ml but ranged between 100 and 180 ml depending on cell density"

Line 170 – It is unfortunate that cell volume estimates of diatoms counted were not measured as this can vary substantially for a given species. It should therefore be noted that these C biomass estimates be taken with extreme caution.
The following line will be modified as such: "Unfortunately the cellular sizes were not measured during diatoms cell counts, thus diatoms were converted to C using average size data compiled for each species from a global 171 ocean database (Leblanc et al., 2012). Results should therefore only meant to present relative evolution of diatoms during the main phases of the experiment and should be interpreted with caution".

Line 218 – : : :were comprised of between: : :
I believe this typo was corrected in the proof reading, it does no longer appear.

Line 347 – It is surprising there is little discussion of Si limitation of diatom growth. Clearly with Si concentrations below 2 uM, this would favor very lightly silicified species (such as C. closterium) or non-diatom phytoplankton groups. I would guess that after PO4 addition, Si is a major regulator of diatom growth (as well as possibly Fe).

Line 400 – Is there evidence to support this hypothesis within the scientific literature? Why would *C. closterium* have such a higher NH4 affinity. It's likely more related to their low Si requirements relative to other diatoms.

My answer to both comments : It is true DSi was on average $1.5 \pm 0.4$ µM which is typical of some tropical low Si waters, and indeed we found the characteristic assemblages of warm water lightly silicified species. I believe that this is overall the case in these waters, and that it is not the DSi concentrations, relatively stable during the experiment that triggered the change towards lightly silicified species, they were all here from the start. However, *C. closterium* is particular in that it is often observed associated to diazotrophs (which was again the case of a recent cruise carried out last year by the same group between Nouméa and Tahiti). Either their nutrient kinetic constants make them best suited to exploit any release of N from diazotrohs, or they have a natural biological association with this group. DFe in the lagoon waters were not measured and could not be handled in trace metal clean way during this experiment, we believe it can hardly constitute a limiting factor for growth inside the mesocosms and that close to the islands.

Line 409 – It's also very likely that these dinoflagelates were mixotrophic. Gyrodinium/ Gymnodinium are well known to exhibit heterotrophy within low nutrient environments when this mode is more favorable.

True. We can modifiy this sentence "It is however possible that dinoflagellates growth may have been stimulated by DDN, but that their biomass was kept unchanged by subsequent grazing." by "It is however possible that dinoflagellates growth may have been stimulated by DDN, but that their biomass was kept unchanged by subsequent grazing, or that their mixotrophic regime allowed them to exploit changes in the dissolved organic pool or go over to phagocytosis (Jeong et al. 2010)"

Jeong, Hae Jin et al. 2010 Growth, feeding and ecological roles of mixotrophic and heterotrophic dinoflagellates in marine planktonic food webs http://dx.doi.org/10.1007/s12601-010-0007-2

Line 463 – Indeed, this might be the case for a number of phytoplankton groups and not just Synechococcus.

Maybe, but the gene expression analyses for the prokaryotic community in Pfreundt et al, this issue-second paper – figure 6, clearly shows that SYN is increasing significantly gene expression for $NH_4$ transporter, while PRO is not at all. But yes other non targeted group could as well, but we have no data to document it either way. Similarly, in this transcriptomic study an increase of sulfolipid gene expression was observed for SYN, and much less for others groups, which again argues for an an adaptative advantage of SYN which decreased its cellular P quota and thus, P demand.

[Figure]

Line 515 – remove "to "in benefited to the entire: : :
OK
Line 517 – observed by, not on. OK

[Figure]

**Figure 7.** Summary of the simplified pathways of N transfer in the first trophic level of the food web and the potential impact on the sinking POC flux at the height of the UCYN-C bloom in the VAHINE mesocosm experiment.

Line 518 – More precisely, what are these clear implications for the efficiency of C export by DDN? What groups were exported?

This synthetic figure from Bonnet et al this issue combines trap data, nifH analyses, $^{15}N$ incubations and nanosims data and illustrates the main results. She shows that the labeled $^{15}N_2$ triggers the UCYN-C bloom, which contributes directly up to 22% of POC in traps (nifH data) and that DDAs contribute up to 7 % of trap POC but also that the $^{15}N$ is tracked up the food chain with the use of Nanosims during parallel incubations in smaller volumes, and stimulates picopk, diatoms and others (zooplankton as well). But unfortunately a clear final relative % of each group in the sediment traps can not be measured in this experiment, which is why she mentions "Potential indirect export" beneath the non diazotroph groups. I find however delicate to change the last sentence to indicate the 30% of POC export was related to UCYN-C (22%) and DDAs (7%) while 70% was from "other groups", since these data are not presented in our paper. But some better indication of this can be added in the discussion section, with reference to Bonnet et al.

Figure 5 – Missing M labels of panels to be consistent with other figures.
Noted

**Anonymous Referee #3**

This papers describes part of a very interesting and exhausitve study about phytoplankton succession and phenology during a mesocosm experiment in LNLC waters. The presented data on floristic identification and pigment analysis is extremely well presented and documented. However, being part of a more exhaustive study containing important complementary data (nutrients, biological processes as production, molecular biology, other diazotroph identifications, TEP concentrations, sediment traps...), it seems very difficult to have an good view of the overall work and results from this study. For instance, there is a clear gap between the "results" section (phytoplankton and pigment results) and the "discussion" section where all other data is combined but not always presented. Most of these other results are described in parallel papers.

It seems to me that this work clearly deserves to be published. However, results shoud be re-structured differently in order to give a broad and complete view and understanding of phytoplankton /ecosystem functionning. I therefore suggest the autors to review the structure of the paper by presenting the required database and information on complementary and essential measured data.

We understand all reviewer's remarks regarding division of data for this specific paper. This paper was one of the last finished, and most papers had a clear focus, set of questions and specific methodologies used ($^{15}N_2$ incubations, Nanosims labelling, qPCR, gene expressions etc…). The diazotrophic community being the main focus, these organisms were already described in two other papers using molecular techniques, with large methodological sections devoted to description of these techniques. We felt among co-authors that it was important the non diazotrophic community was described as well to help understand the evolution of the main biogeochemical stocks and fluxes not necessarily dominated by the diazotrophs themselves. Knapp et al (this issue) show that not all exported N originates from $^{15}N_2$ fixation, and Van Wambeke et al. (this issue) further states that $N_2$ fixation fluxes can only support a small part of the bacterial production, so other sources of N are to be considered.

The main issue here with including the diazotroph community in this paper, apart from being redundant with other papers, was that there are no direct cell counts of these groups. Data entirely relies on the qPCR data and nifH gene expression. I agree it is very unfortunate these organisms were not enumerated directly, but I suspect techniques for doing this are quite difficult to use and that molecular data was quicker to obtain and sufficed to answer the main objectives of the project. This being said, we agree among co-authors, and if the editors agree, to move up the diazotroph community data from supplement to results section, and add some short methodological section referring to Turk-Kubo et al's paper. However this data set is based on nifH gene abundance and not on cell number, so in any case they will NOT be directly comparable to the other groups data presented here which are all in cells $L^{-1}$. Conversion from nifH copies to cell number is at present not reliable as number of nifH copies per cell can vary from 1 to 32 for instance just taking the number of *Richelia* symbionts found in the diatom *Rhizosolenia clevei*. If needed, we can also add short sections in M&M referring to nutrient, primary production, and $^{15}N_2$ fixation fluxes with reference to the other papers where these data are lengthily presented, to avoid this gap between M&M / results / and discussion. My main concern which prevented me from adding diazotrophs and other biogeochemical fluxes was that it was all already presented elsewhere, but again, if editors agree, we can easily harmonize M&M and results with our discussion. Our results are also presented in a very synthetic way, as boxplots for the main periods, so the data would not be presented in the same graphical way than in the other papers.

---

## Editor Decision (ED1)

Review of revised manuscript by K. Leblanc et al., "Phytoplankton community structure in the VAHINE MESOCOSM experiment"

Leblanc and colleagues present a revised version of their manuscript on floristic results from a LNLC mesocosm experiment in Noumea designed to stimulate diazotrophy and follow the transfer of newly fixed N through the ecosystem. The original manuscript, though well-written and interesting, was missing key datasets and methods, which made it impossible to consider as a standalone paper. The authors have addressed those deficiencies adequately and I now recommend publication after minor revisions. I do not think another round of review is necessary; however, I ask the Editor to ensure that these remaining minor points are adequately considered by the authors:

**Title:** Shouldn't "MESCOCOSM" be all lower case? It doesn't appear to be an acronym.

Line 20: "phosphate" should be "phosphorus".

Line 21: "Initially, the diazotrophic community was...".

Line 23: Add comma before "which".

Line 24: Delete comma before "that".

Line 39:  $N_2$  is "a" major source, not "the" major source of new N to the ocean.

Line 49: "diazotrophs" (plural).

Line 123: The glycerol uncoupling method for PE is an in vivo method, not an extractive method.

Line 182: What is "gentle" peristaltic pumping? Come on now...

**Line 208:** Was the enrichment of the 15N stock assessed by MIMS? This point needs to be made clear. In practice, the measured enrichment is typically lower than that expected from calculations, due to air  $N_2$  contamination during preparation.

Line 212: Superscript "15".

Line 216: Need to state the chemical form of the added 14C.

**Lines 232-238:** The glycerol uncoupling method was developed for *Synechococcus* (very small cells); because it is not extractive, it is possible that there are packaging effects associated with larger PE-containing organisms such as *Trichodesmium* and DDAs. To my knowledge, these potential artifacts have never been characterized. That is, the method is likely only semi-quantitative for the larger organisms. This point should be made here in the context of PE changes over time.

Line 242: "whose" not "which".

Line 250: "In contrast to..." not "Contrary to...".

Line 266: "nano-phytoeukaryote" (singular).

Line 266: Delete "comprised".

Line 275: "whose" not "which".

Line 349: "diazotroph" (singular).

**Line 352**: I don't think it is correct to say that these rates were "among the highest ever reported". Certainly many terrestrial and freshwater systems exhibit higher N2 fixation rates, and as far as marine systems go I suspect in the Baltic, for example, rates can be extremely high. Perhaps these rates are among the highest ever reported from this lagoon. Be specific.

Line 429: Add "of" before "DDN".

**Lines 449-453:** Could the increase in *C. closterium* be due to a mesocosm wall effect? You do mention that they can be common in benthic environments.

Line 492: Change "clearly" to "likely".

Lines 497-500: If you are going to tell me that "Clear differences" between the mesocosms and the lagoon exist, you'll need to state the statistical test and P-value. Otherwise, use less strong wording. Figures 3, 12, 13 and 14: I'm not a fan of the funny European habit of using commas in place of decimal points; either way, you need to be consistent throughout.

---

## Author Response (AR2)

Review of revised manuscript by K. Leblanc et al., "Phytoplankton community structure in the VAHINE MESOCOSM experiment"

Leblanc and colleagues present a revised version of their manuscript on floristic results from a LNLC mesocosm experiment in Noumea designed to stimulate diazotrophy and follow the transfer of newly fixed N through the ecosystem. The original manuscript, though well-written and interesting, was missing key datasets and methods, which made it impossible to consider as a standalone paper. The authors have addressed those deficiencies adequately and I now recommend publication after minor revisions. I do not think another round of review is necessary; however, I ask the Editor to ensure that these remaining minor points are adequately considered by the authors:

**Title:** Shouldn't "MESCOCOSM" be all lower case? It doesn't appear to be an acronym. OK

Line 20: "phosphate" should be "phosphorus". OK
Line 21: "Initially, the diazotrophic community was...". OK
Line 23: Add comma before "which". OK
Line 24: Delete comma before "that". OK
Line 39: N2 is "a" major source, not "the" major source of new N to the ocean. OK
Line 49: "diazotrophs" (plural). OK

Line 123: The glycerol uncoupling method for PE is an in vivo method, not an extractive method.

In answer to this comment + the one on line 232-238, the text in the method section was modified as follows :

"Water samples (4.5 L) were filtered onto 0.4  $\mu$ m Nucleopore polycarbonate membrane filters (47 mm diameter) and immediately frozen in liquid nitrogen until analysis. In the laboratory, particles retained on the filter were resuspended in a 4 mL glycerol-phosphate mixture (50/50) after vigorous shaking, according to the *in vivo* method (Wyman, 1992). The PE fluorescence excitation spectra were recorded between 450 and 580 nm (emission fixed at 605 nm), using a Perkin Elmer LS55 spectrofluorometer and emission and excitation slit widths adjusted to 5 and 10 nm, respectively (Neveux et al. , 2009). As this method was developped for small *Synechococcus* cells, potential packaging effect could occur when measuring PE in larger cells such as *Trichodesmium*, but this remains to be documented. Estimates of phycoerythrin were obtained from the area below the fluorescence excitation curve, after filter blank subtraction. PE analyses were made only at 6 m-depth in the three mesocosms and in lagoon waters.

**Line 182:** What is "gentle" peristaltic pumping? Come on now... OK "gentle" was removed

**Line 208:** Was the enrichment of the  $_{15}N$  stock assessed by MIMS? This point needs to be made clear. In practice, the measured enrichment is typically lower than that expected from calculations, due to air N2 contamination during preparation.

Yes it was. Following text was added :

"The 15N enrichment of the N2 pool was measured by Membrane Inlet Mass Spectrometer according to Kana et al. (1994) and was found to be 2.4  $\pm$  0.2 atom%. "

Kana, T. M., Darkangelo, C., Hunt, M. D., Oldham, J. B., Bennett, G. E., and Cornwell, J. C.: Membrane Inlet Mass Spectrometer for Rapid High-Precision Determination of N2, O2, and Ar in Environmental Water Samples, Analytical Chemistry, 66 (23), 4166-4170, doi: 10.1021/ac00095a009, 1994.

Line 212: Superscript "15". OK

Line 216: Need to state the chemical form of the added 14C. OK

**Lines 232-238:** The glycerol uncoupling method was developed for *Synechococcus* (very small cells); because it is not extractive, it is possible that there are packaging effects associated with larger PE containing organisms such as *Trichodesmium* and DDAs. To my knowledge, these potential artifacts have never been characterized. That is, the method is likely only semi-quantitative for the larger organisms. This point should be made here in the context of PE changes over time.

Line 242: "whose" not "which". OK

Line 250: "In contrast to ... " not "Contrary to ... ". OK

Line 266: "nano-phytoeukaryote" (singular). OK

Line 266: Delete "comprised". OK

Line 275: "whose" not "which". OK Line 349: "diazotroph" (singular). OK

**Line 352**: I don't think it is correct to say that these rates were "among the highest ever reported". Certainly many terrestrial and freshwater systems exhibit higher N2 fixation rates, and as far as marine systems go I suspect in the Baltic, for example, rates can be extremely high. Perhaps these rates are among the highest ever reported from this lagoon. Be specific.

Following text was added "ever reported for oceanic systems"

Line 429: Add "of" before "DDN". OK

**Lines 449-453:** Could the increase in *C. closterium* be due to a mesocosm wall effect? You do mention that they can be common in benthic environments.

Cylindrotheca closterium could in effect grow on cell wall, but similarly to all other pennates. Yet it is more likely that they benefited from the locally modified conditions and synchronous increase with UCYN-C. Cylindrotheca closterium is a most opportunistic diatom and often blooms in mesocosms / microcosms experiments. Even if it grew on cell walls (which was not verified under this experiment) it would still grow there because of beneficial bottom-up conditions in the water tanks, so I am not sure adding a comment without any proof either way would make this point any clearer.

Line 492: Change "clearly" to "likely". OK

Lines 497-500: If you are going to tell me that "Clear differences" between the mesocosms and the

lagoon exist, you'll need to state the statistical test and P-value. Otherwise, use less strong wording. OK

**Figures 3, 12, 13 and 14:** I'm not a fan of the funny European habit of using commas in place of decimal points; either way, you need to be consistent throughout. OK decimal points were corrected in the Figure file.